# FIRING-NET: A FILTERED FEATURE RECYCLING NETWORK FOR SPEECH ENHANCEMENT

**Xinmeng Xu**[1]**, Jizhen Li**[1]**, Yiqun Zhang**[1]**, Yong Luo**[1]**, Yuhong Yang**[1,2,*] **& Weiping Tu**[1,2,*]

[1]NERCMS, School of Computer Science, Wuhan University, China
[2]Hubei Key Laboratory of Multimedia and Network Communication Engineering,
Wuhan University, China
`{xuxinmeng, yangyuhong, tuweiping}@whu.edu.cn`

## ABSTRACT

Current deep neural networks for speech enhancement (SE) aim to minimize the distance between the output signal and the clean target by filtering out noise features from input features. However, when noise and speech components are highly similar, SE models struggle to learn effective discrimination patterns. To address this challenge, we propose a Filter-Recycle-Interguide framework termed **FI**lter-**R**ecycle-**IN**ter**G**uide **NET**work (FIRING-Net) for SE, which filters the input features to extract target features and recycles the filtered-out features as non-target features. These two feature sets then guide each other to refine the features, leading to the aggregation of speech information within the target features and noise information within the non-target features. The proposed FIRING-Net mainly consists of a Local Module (LM) and a Global Module (GM). The LM uses outputs of the speech extraction network as target features and the residual between input and output as non-target features. The GM leverages the energy distribution of the self-attention map to extract target and non-target features guided by the highest and lowest energy regions. Both LM and GM include interaction modules to leverage the two feature sets in an inter-guided manner for collecting speech from non-target features and filtering out noise from target features. Experiments confirm the effectiveness of the Filter-Recycle-Interguide framework. Additionally, FIRING-Net achieves a good balance between SE performance and computational efficiency, outperforming other comparable models across various signal-to-noise ratio levels and noise environments.

## 1 INTRODUCTION

Speech enhancement (SE) aims to separate speech from background interference signals (Xu et al., 2013). It is a fundamental speech processing problem and has been frequently used as the pre-processor of several acoustic tasks, such as speech recognition (Peng et al., 2022), speaker verification (Rao et al., 2019), speaker diarization (Sell et al., 2018), etc. Traditional SE approaches, such as Wiener filtering (Chen et al., 2006), spectral subtraction (Vaseghi, 2008), and principle component analysis (Srinivasarao & Ghanekar, 2020), assume that the noises belong to stationary signals, which are significantly different from the speech signal. However, this assumption usually cannot be satisfied in practice and thus these approaches often fail in real-world applications.

Recently, deep networks have shown their promising performance on SE, even under highly non-stationary noise environments. Deep learning-based SE methods train SE networks on extensive noisy-clean pairs to learn to distinguish between speech and noise, aiming to effectively remove noise components (Sell et al., 2018; Xu et al., 2013). For example, we can utilize the convolutional operations to extract local speech features (Park & Lee, 2017; Pandey & Wang, 2019), the recurrent neural networks or transformers to capture global speech features (Sun et al., 2017; Kim et al., 2020), and integrating both techniques can enhance internal feature differences within speech (Abdulatif et al., 2024; Lu et al., 2023; Xu et al., 2023). Despite significant improvements, issues of speech distortion and residual noise persist (Jo & Yoo, 2010; Xia et al., 2020; Wakabayashi et al., 2018).

---

*Corresponding Authors.

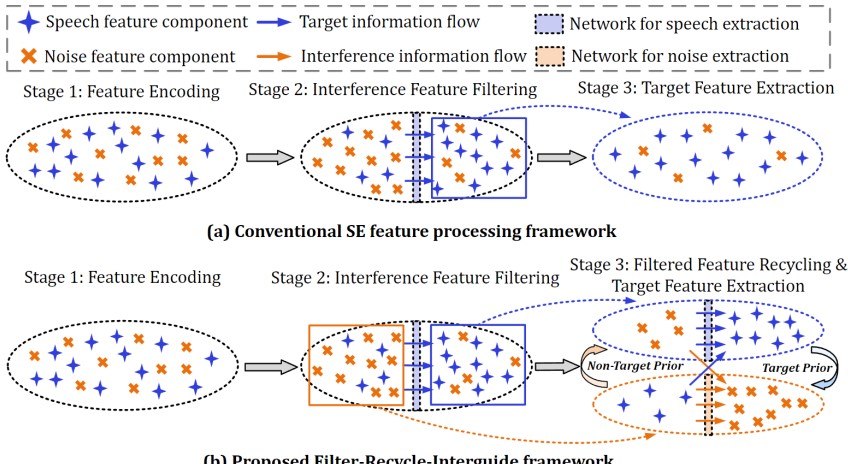

Figure 1: Comparison of (a) the conventional SE framework and (b) the proposed Filter-Recycle-Interguide framework. (a) The conventional framework filters input features to remove noise but risks overestimating or underestimating noise. (b) The proposed framework recycles filtered features and processes them interactively with output features. This approach leverages the speech-dominant output and noise-dominant filtered features for mutual guidance, enhancing feature separation.

That is, they remain ineffective against noise containing highly similar acoustic events, such as speech from non-target speakers or echoes reflected from walls (Hu et al., 2023; Zheng et al., 2021). The primary reason is the often indistinct differences between speech and noise components. As shown in Figure 1(a), SE networks struggle to learn discriminative patterns between highly similar speech and noise components, leading to the overestimation or underestimation of noise information.

Some recent solutions use noise as part of the supervision to enable SE networks to establish a dual mapping from mixed signals to both speech and noise signals. A representative work is the dual-branch network structure, where the two branches aim to model the speech and noise signal, respectively, and an interacting operation is designed to explore the correlation between speech and noise features in a noisy mixture (Zheng et al., 2021; Zhao et al., 2022; Li et al., 2024). This method interactively leverages the features of speech and noise signals and using both as supervision. The goal is to ensure the relative independence of speech and noise signals in the feature space, and thereby enabling the network to extract discriminative patterns for each signal. However, due to the dynamic and highly random nature of noise, completely predicting a noise signal is infeasible (Ortega-García & González-Rodríguez, 1996). Consequently, its performance still heavily relies on the accuracy of the speech signal prediction branch, thereby diminishing the primary purpose of predicting noise signals.

To remedy this drawback, we design a Filter-Recycle-Interguide (FRI) strategy to extract more discriminative patterns between speech and noise within the mixture input. This is achieved by: (1) avoiding overemphasis on exploring feature differences within speech and (2) achieving more robust noise feature extraction for interaction with speech features without directly predicting the noise signal. As shown in Figure 1(b), our method consists of three main steps: (1) Processing input features through network modules to extract target features, similar to conventional SE models; (2) Re-collecting and redefining filtered-out features as non-target features; (3) Guiding and interacting target and non-target features, where target features, mainly composed of speech components, integrate speech from non-target features and remove noise, while non-target features, predominantly containing noise, assist in refining the target features.

By incorporating the designed FRI strategy into deep neural networks, we obtain the proposed **FI**lter-**R**ecycle-**IN**ter**G**uide **NET**work (FIRING-Net). IRING-Net integrates local and global perspectives through the Local Module (LM) and Global Module (GM) to perform Filtering and Recycling operations on noisy speech features. The LM refines local features using residual calculations between input and output features, inspired by back-projection (Afouras et al., 2018), to extract non-target features. The GM addresses global dependencies by leveraging energy distributions in

self-attention maps and applying a Top-K mechanism (Chen et al., 2023b) for non-target feature refinement. To address the overlap and similarity between speech and noise, both modules adopt a two-stage coarse-to-fine framework, enabling precise separation of target and non-target features, ensuring speech-dominant and noise-dominant characteristics are accurately distinguished. Additionally, FIRING-Net considers the distinct roles of encoder and decoder features: the encoder primarily contains noise, while the decoder focuses on speech reconstruction. During the interguide stage, tailored interaction modules are employed for each, facilitating effective noise removal in the encoder and enhancing the completeness of reconstructed speech in the decoder. This comprehensive design ensures robust speech separation and reconstruction, even under challenging noise conditions.

Our model is trained on DNS Challenge 2021 (Reddy et al., 2021) and is evaluated on two public datasets (WSJ0-SI 84 + NOISX-92 (Paul & Baker, 1992; Varga & Steeneken, 1993) and AVSpeech + AudioSet (Ephrat et al., 2018; Gemmeke et al., 2017)). The extensive results demonstrate the efficacy of the proposed method. Specifically, under the babble noise environment, it achieves a $3.03\%$ relative improvement in terms of PESQ and a $6\%$ dB gain in terms of SI-SNRi compared to the most competitive MP-SENet (Lu et al., 2023).

To summarize, the main contributions of this paper are:

- We design a Filter-Recycle-Interguide SE strategy, which enables mutual refinement between retained and filtered-out features, i.e., effectively eliminating noise from the retained features while recovering speech information from the filtered-out features.

- We propose FIRING-Net for SE, comprising two key components: LM and GM. Both modules employ a two-stage coarse-to-fine framework, effectively separating target and non-target features, ensuring that speech-dominant and noise-dominant characteristics are accurately captured.

- We propose two distinct interaction modules employed for the encoder and decoder, enabling comprehensive noise removal in the encoder and enhancing the completeness of reconstructed speech in the decoder.

## 2 RELATED WORKS

The integration of deep learning has significantly enhanced SE performance. Recently, convolutional neural networks (CNNs) have been employed in SE tasks with notable success, as they effectively capture implicit information within the speech signal and manage small shifts in the time-frequency (TF) domain of speech features, thus adapting to varying speaker identities and acoustic environments (Park & Lee, 2017; Fu et al., 2017). However, CNNs are somewhat limited in modeling broader dependencies in low-level features (O'shea & Nash, 2015). To address this issue, many approaches have turned to transformers (Vaswani et al., 2017) to replace CNNs, enabling the capture of global information in waveforms or spectrograms (Kim et al., 2020; Dang et al., 2022a). Furthermore, several studies have sought to integrate CNNs and transformers, thereby harnessing both local and global information (Abdulatif et al., 2024; Lu et al., 2023). However, deep learning-based SE methods primarily focus on extracting speech features based on speech characteristics alone, leading to speech distortion or residual noise in the enhanced output when speech and noise components in the deep latent space are highly similar.

Rather than solely focusing on estimating target speech, some SE methods improve the performance by building noise models to account for noise prior. For instance, certain approaches (Odelowo & Anderson, 2017; Liu et al., 2021) implement spectral subtraction using deep neural networks (DNNs) through a two-stage process: noise signal estimation followed by speech signal recovery. Although outperform traditional signal processing techniques in modeling structured noise (Ortega-García & González-Rodríguez, 1996), DNNs' generalization capability remains limited. To enhance noise modeling accuracy, advanced methods (Zheng et al., 2021; Zhao et al., 2022; Li et al., 2024) introduce a two-branch framework to predict both speech and noise simultaneously, incorporating interaction modules at various layers based on the correlations between predicted speech and the residual signal. However, this approach still struggles with the unpredictability of noise, as its success largely depends on the accuracy of the speech prediction branch.

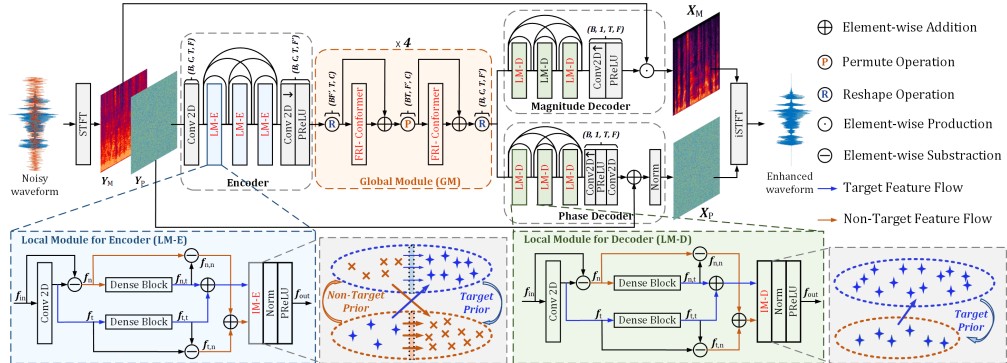

Figure 2: Overview of our FIRING-Net, which is an encoder-decoder architecture that leverages the Filter-Recycle-Interguide strategy for SE by dividing input features into target (speech) and non-target (noise) components. FIRING-Net integrates the LM and GM to implement the FRI strategy through distinct target and non-target feature separation methods. The LM extracts non-target features by calculating the error between input and output, i.e., target features, while the GM leverages self-attention energy distributions to separate target and non-target features. In the encoder, where non-target features are noise-dominant, the IM-E facilitates reciprocal extraction of speech and noise from target and non-target features. In the decoder, where non-target features are speech-dominant, the IM-D ensures comprehensive integration of target and non-target features.

## 3  METHOD

Figure 2 is an overall illustration of our method. The noisy signal $\mathbf{y} \in \mathbb{R}^{1 \times L}$ undergoes a short-time Fourier transform (STFT) to produce a complex spectrogram $\mathbf{Y} \in \mathbb{R}^{2 \times T \times F}$. We extract the magnitude spectrogram $\mathbf{Y}_M \in \mathbb{R}^{1 \times T \times F}$ and the wrapped phase spectrogram $\mathbf{Y}_P \in \mathbb{R}^{1 \times T \times F}$, where $T$ and $F$ denote time and frequency. Before processing with FIRING-Net, a power-law compression (Wisdom et al., 2019) is applied to $\mathbf{Y}_M$ using a compression factor of 0.3 (Braun & Tashev, 2021), to enhance alignment with human auditory perception (Lee et al., 2018; Wilson et al., 2018). The resulting compressed magnitude spectrogram $\mathbf{Y}_M^c$ is then concatenated with $\mathbf{Y}_P$ and used as the input for FIRING-Net.

**Encoder.** Given the input feature $\mathbf{Y}_{in} \in \mathbb{R}^{2 \times T \times F}$, the encoder includes two convolution blocks, together with three densely connected Local Modules for encoder (LM-Es) positioned between the two convolutional blocks. The first convolution block maps $\mathbf{Y}_{in}$ to an intermediate space with $C$ channels. Three LM-Es aim to extract more completed speech information and accurately suppress noise by using the Filter-Recycle-Interguide framework (see Sec 3.1), while dense connections combine features from all layers to capture different levels of detail. The final convolution block decreases the frequency dimension to $F'$ to reduce complexity. Subsequently, the encoder output is then processed by $N$ Global Modules (GMs), with $N$ set to 4, as described in Sec 3.2. The architecture, inspired by TF-Conformer (Abdulatif et al., 2024; Lu et al., 2023), alternately captures time and frequency dependencies within the Filter-Recycle-Interguide framework.

**Magnitude and Phase Decoders.** The decoder processes the output from the $N$ GMs in a decoupled manner, involving two branches: the magnitude decoder and the phase decoder. The former is designed to predict a mask, which is then element-wise multiplied by the input magnitude spectrogram to enhance the signal. In contrast, the phase decoder predicts a residual that refines the noisy phase to improve phase accuracy (Afouras et al., 2018). Both decoders contain three densely connected LMs for decoder (LM-Ds). In both branches, a deconvolutional block is employed to upsample the frequency dimension back to $F$, while reducing the channel number to 1. For the magnitude decoder, a PReLU activation function is used to predict the final mask, allowing it to learn different slopes for each frequency band (Abdulatif et al., 2024). In the phase decoder, the predicted phase residual is added to the noisy phase and then normalized to produce a clean phase prediction (Afouras et al., 2018).

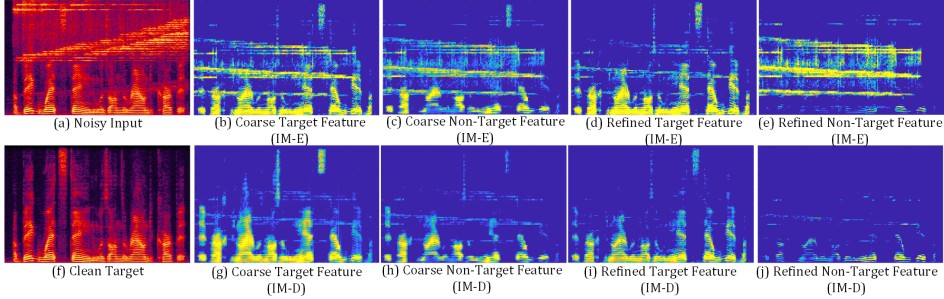

Figure 3: Visualizations of coarse and refined target and non-target features, i.e., $\mathbf{f}_t$, $\mathbf{f}_n$, $\hat{\mathbf{f}}_t$, and $\hat{\mathbf{f}}_n$, extracted by LM-E and LM-D, along with their corresponding spectrograms of noisy mixtures and clean target speech.

## 3.1 LOCAL MODULE

We replace the conventional convolutional modules in the Encoder and Magnitude and Phase Decoders with Local Module for Encoder (LM-E) and Local Module for Decoder (LM-D). As shown in Figure 2, LM-E and LM-D have identical structures, with the only difference being that LM-E contains an Interaction Module specifically designed for the Encoder part (IM-E), while LM-D includes an Interaction Module tailored for the Decoder part (IM-D). The LM coarsely produce a target feature $\mathbf{f}_t$ processed from the input feature $\mathbf{f}_{in}$ by applying a convolutional layer with normalization and filter out the non-target feature $\mathbf{f}_n$,

$$\mathbf{f}_t = \mathcal{H}_{conv}(\mathbf{f}_{in}), \qquad \mathbf{f}_n = \mathbf{f}_{in} - \mathbf{f}_t, \tag{1}$$

where $\mathcal{H}_{conv}(\cdot)$ denotes the convolutional operation.

A simple convolutional layer, due to its limited capacity, only provides a coarse separation of target and non-target features within the $\mathbf{f}_{in}$. Consequently, we further refine both the target and non-target features separately. For the non-target features, we use a dilated DenseNet (Pandey & Wang, 2020) to further extract the ignored target features $\mathbf{f}_{n,t}$ and calculate the residual between $\mathbf{f}_{n,t}$ and $\mathbf{f}_n$ to obtain more purified non-target feature $\mathbf{f}_{n,n}$. Similarly, we apply the same process to the target features, resulting in more purified target feature $\mathbf{f}_{t,t}$ as well as contributing to the refinement of non-target feature $\mathbf{f}_{t,n}$,

$$\mathbf{f}_{n,t} = \mathcal{H}_{dense}(\mathbf{f}_n), \qquad \mathbf{f}_{n,n} = \mathbf{f}_n - \mathbf{f}_{n,t}, \tag{2}$$
$$\mathbf{f}_{t,t} = \mathcal{H}_{dense}(\mathbf{f}_t), \qquad \mathbf{f}_{t,n} = \mathbf{f}_t - \mathbf{f}_{t,t}, \tag{3}$$

where $\mathcal{H}_{dense}(\cdot)$ denotes the dilated DenseNet that contains three convolution blocks with dense connections, the dilation factors of each block are set to $\{1, 2, 4\}$. We aggregate all target and non-target features through an addition operation to obtain the refined target features $\hat{\mathbf{f}}_t = \mathbf{f}_{t,t} + \mathbf{f}_{n,t}$ and non-target feature $\hat{\mathbf{f}}_n = \mathbf{f}_{t,n} + \mathbf{f}_{n,n}$, respectively.

The purpose of the Interaction Module is to enable mutual guidance between target and non-target features, allowing for a more refined extraction of speech information from non-target features to supplement the target features, while filtering out noise information from the target features. However, since the information contained in non-target features differs between the Encoder and Decoder, we designed IM-E and IM-D to specifically handle the target and non-target features in LM-E and LM-D, respectively.

**LM-E**: In the Encoder, input features typically contain more noise components. As shown in Figure 3(c), the non-target features extracted by LM-E are predominantly noise. The role of IM-E is to extract speech components from these non-target features and filter out noise from the target features. As depicted in Figure 4(a), IM-E concatenates $\hat{\mathbf{f}}_t$ and $\hat{\mathbf{f}}_n$ and feeds them into a convolutional module to generate a mask $\mathbf{M}_n$. This mask identifies the preserved areas of non-target features to extract speech information, which is then added to $\hat{\mathbf{f}}_t$ to produce a more speech information completed target feature $\mathbf{f}'_t$. Additionally, $|Sigmoid(\hat{\mathbf{f}}_n) - \mathbf{M}_n|$ isolates purer noise information from the non-target features (The reason for using "$|Sigmoid(\hat{\mathbf{f}}_n) - \mathbf{M}_n|$" instead of "$1 - \mathbf{M}_n$" is explained in

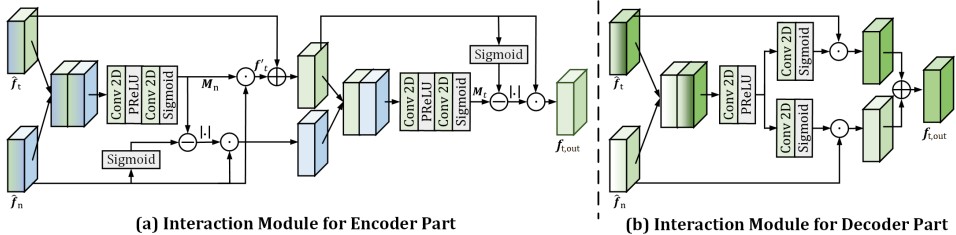

Figure 4: Network architecture of the Interaction Module (IM): (a) The IM in the encoder extracts speech from non-target features guided by target features and removes noise from target features guided by non-target features. (b) The IM in the decoder, where noise is largely suppressed, refines discarded speech information from non-target features using target feature guidance for a more complete speech reconstruction.

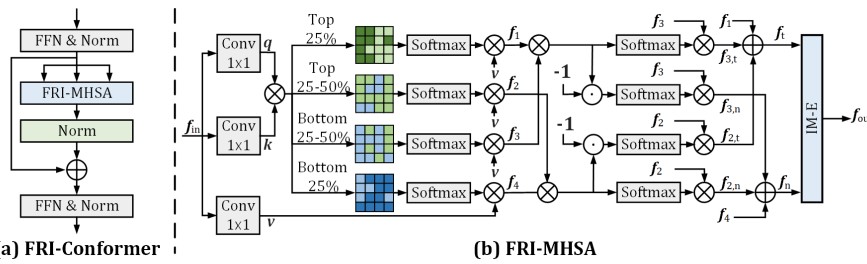

Figure 5: Network architecture of the Filter-Recycle-Interguide Conformer (FRI-Conformer) in the global module, featuring the crucial Filter-Recycle-Interguide Multi-Head Self-Attention (FRI-MHSA). FRI-MHSA classifies input features into four parts based on attention map energy distribution, using the highest and lowest energy regions to guide speech extraction from mid-low energy features and noise extraction from mid-high energy features.

the Appendix B.3), which is combined with $\mathbf{f}'_t$ to guide noise extraction from $\mathbf{f}'_t$, before feeding into a convolutional module to generate a mask $\mathbf{M}_t$ that is subtracted by "$Sigmoid(\hat{\mathbf{f}}_t)$", and the absolute value is taken afterward, i.e., $|Sigmoid(\hat{\mathbf{f}}_t) - \mathbf{M}_t|$, to filter out remaining noise.

**LM-D**: In the Decoder, input features typically contain minimal or no noise components. As illustrated in Figure 3(f), non-target information often contains more speech components that are difficult to use for target speech reconstruction. Therefore, IM-D ensures mutual guidance between target and non-target features, allowing speech information from non-target features to contribute effectively to the reconstruction of the target speech, thus reducing speech distortion. As shown in Figure 4(b), IM-D concatenates $\hat{\mathbf{f}}_t$ and $\hat{\mathbf{f}}_n$ and processes them through a convolutional module to generate an intermediate feature. This intermediate feature is then processed by two separate convolutional modules to produce masks for $\hat{\mathbf{f}}_t$ and $\hat{\mathbf{f}}_n$. The masked $\hat{\mathbf{f}}_t$ and $\hat{\mathbf{f}}_n$ are then combined to produce the output target feature with enhanced speech information.

## 3.2 GLOBAL MODULE

The GM is designed to overcome the limitations of the LM, which can only extract target and non-target features within a limited receptive field. By integrating the long-range contextual modeling capabilities of Transformers (Vaswani et al., 2017), the Global Module facilitates the extraction of these features from a global perspective of the speech signal. As depicted in Figure 2, the GM adopts a dual-path attention-based structure (Abdulatif et al., 2024; Wang et al., 2021) and employs two FRI-Conformer blocks in sequence. The first stage captures time dependencies with an input shape of $BF' \times T \times C$, where $B$ represents the batch size, while the second stage captures frequency dependencies with an input shape of $BT \times F' \times C$. As depicted in Figure 5(a), each FRI-Conformer employs two half-step feedforward networks (FFNs) with a Filter-Recycle-Interguide Multi-head Self-attention (FRI-MHSA) module in between.

The design of FRI-MHSA is based on the energy distribution in the attention feature map, where high-energy regions guide the extraction of speech features in low-energy regions, and low-energy regions guide the extraction of noise features in high-energy regions. Given a query $\mathbf{q}$, key $\mathbf{k}$, and value $\mathbf{v}$, the output of dot-product attention is generally formulated as:

$$\text{Att}(\mathbf{q}, \mathbf{k}, \mathbf{v}) = \text{softmax}(\mathbf{q}\mathbf{k}^\top)\mathbf{v}. \tag{4}$$

In our work, FRI-MHSA segments the feature map obtained from $\mathbf{q}\mathbf{k}^\top$ based on energy levels, extracting the top 25% highest energy regions, the upper-middle 25%, the lower-middle 25%, and the bottom 25% lowest energy regions. We draw inspiration from the top-k operation (Chen et al., 2023b; Xiao et al., 2024) to implement this extraction and assign an infinitesimal value to the unextracted portions of each feature map. Subsequently, we apply softmax to each of the four feature maps, generating four masks that extract four types of features from $\mathbf{v}$. The mask generated from the top 25% energy feature map extracts target features $\mathbf{f}_1$ from $\mathbf{v}$, primarily containing speech information, which guides the extraction of speech information from the lower-middle 25% energy feature map $\mathbf{f}_3$. Conversely, the mask corresponding to the bottom 25% energy feature map extracts non-target features $\mathbf{f}_4$ from $\mathbf{v}$, primarily containing noise information, which in turn guides the extraction of noise information from the upper-middle 25% energy feature map $\mathbf{f}_2$. This guided extraction method is based on calculating the cross-similarity between two types of features through matrix multiplication, allowing the extraction of the parts from the guided features that are most similar to the guiding features. Therefore, the above process can be expressed as:

$$\mathbf{f}_{3,t} = \text{softmax}(\mathbf{f}_1\mathbf{f}_3^\top)\mathbf{f}_3, \qquad \mathbf{f}_{3,n} = \text{softmax}(-1 * (\mathbf{f}_1\mathbf{f}_3^\top))\mathbf{f}_3, \tag{5}$$

$$\mathbf{f}_{2,n} = \text{softmax}(\mathbf{f}_4\mathbf{f}_2^\top)\mathbf{f}_2, \qquad \mathbf{f}_{2,t} = \text{softmax}(-1 * (\mathbf{f}_4\mathbf{f}_2^\top))\mathbf{f}_2, \tag{6}$$

where $\mathbf{f}_{2,t}$ and $\mathbf{f}_{2,n}$ represent the target and non-target features extracted from $\mathbf{f}_2$, and $\mathbf{f}_{3,t}$ and $\mathbf{f}_{3,n}$ represent the target and non-target features extracted from $\mathbf{f}_3$. Finally, the target feature is obtained by $\mathbf{f}_t = \mathbf{f}_1 + \mathbf{f}_{2,t} + \mathbf{f}_{3,t}$, while the non-target feature is obtained by $\mathbf{f}_n = \mathbf{f}_4 + \mathbf{f}_{2,n} + \mathbf{f}_{3,n}$. These features, $\mathbf{f}_t$ and $\mathbf{f}_n$ are fed into IM-E for further processing.

# 4 EXPERIMENTAL SETUP

## 4.1 DATASET

We trained FIRING-Net using the Interspeech 2021 DNS Challenge dataset (Reddy et al., 2021), sampled at 16 kHz. A total of 60,000 reverberant speech clips, approximately 500 hours in duration, were generated, with 55,000 clips designated for training and 5,000 for validation. The noise clips were mainly sourced from Audioset (Gemmeke et al., 2017), DEMAND (Thiemann et al., 2013a), and Freesound (Fonseca et al., 2017). During training, the audio was randomly segmented into 4-second clips and processed with randomly selected room impulse responses (RIRs) from OpenSLR26 and OpenSLR28 (Ko et al., 2017) ($T_{60}$ in the range from 0.3s to 1.3s). The noisy speech was created by mixing reverberant speech with noise, with the SNR range set between -5 dB and 5 dB. We selected two datasets as the test sets for performance evaluation under various unknown noise conditions:

**WSJ0-SI 84 + NOISEX-92:** We selected 651 utterances from 8 speakers in the WSJ0-SI 84 dataset (Paul & Baker, 1992). Noise samples were taken from the NOISEX-92 dataset (Varga & Steeneken, 1993), and test mixtures were created by combining these noise samples with the speech at SNR levels of -5 dB, 0 dB, and 5 dB in reverberant conditions, with $T_{60}$ values randomly selected between 0.3s and 1.3s.

**AVSpeech + AudioSet:** The AVSpeech dataset consists of public instructional YouTube videos, from which 3-10s clips were automatically extracted, ensuring that the only audible sound in each clip is from a single speaker (Ephrat et al., 2018). For our experiments, we downloaded 1,199 clips from the test set, utilizing only the audio portions. Four representative noise types: babble, engine, baby cry and laughter, and traffic, from the Audioset dataset (Gemmeke et al., 2017) were introduced, and noisy speech is generated through a weighted linear combination of clean utterances from AVSpeech and noise segments from AudioSet,

$$x_i = s_j^{AVSpeech} + 0.3 \cdot v_k^{AudioSet}, \tag{7}$$

where $s_j^{AVSpeech}$ and $v_k^{AudioSet}$ represent 4-second randomly sampled speech and noise segments, respectively.

Table 1: Comparison with selected baseline models on WSJ0-SI 84+NOISEX-92 with different SNR levels. Bold and underline indicate the best and second-best results.

| Method | Param. | -5 dB | | | 0 dB | | | 5 dB | | |
|---|---|---|---|---|---|---|---|---|---|---|
| | | STOI | PESQ | SI-SNRi | STOI | PESQ | SI-SNRi | STOI | PESQ | SI-SNRi |
| Unprocessed | - | 0.6172 | 1.48 | - | 0.7813 | 1.76 | - | 0.8669 | 1.92 | - |
| PHASEN | 6.41M | 0.7886 | 2.25 | 12.84 | 0.8981 | 2.85 | 11.82 | 0.9238 | 3.09 | 9.45 |
| SN-Net | 8.14M | 0.8157 | 2.31 | 13.18 | 0.9031 | 2.93 | 12.10 | 0.9305 | 3.14 | 10.05 |
| Inter-SubNet | 2.29M | 0.8038 | 2.29 | 13.41 | 0.8968 | 2.89 | 11.95 | 0.9287 | 3.13 | 10.78 |
| CMGAN | 1.83M | 0.8205 | 2.38 | 14.29 | 0.9085 | 2.97 | 12.53 | 0.9324 | 3.17 | 11.65 |
| MP-SENet | 2.05M | 0.8342 | 2.43 | 15.68 | 0.9112 | 3.03 | 13.69 | 0.9384 | 3.21 | 11.74 |
| FIRING-Net | **1.81M** | **0.8414** | **2.51** | **16.11** | **0.9195** | **3.09** | **14.20** | **0.9430** | **3.28** | **12.06** |

Table 2: Comparison with selected baseline models on AVSpeech + AudioSet with different types of noise. Bold and underline indicate the best and second-best results.

| Method | Babble | | | Engine | | | Baby Cry and Laughter | | | Traffic | | |
|---|---|---|---|---|---|---|---|---|---|---|---|---|
| | STOI | PESQ | SI-SNRi | STOI | PESQ | SI-SNRi | STOI | PESQ | SI-SNRi | STOI | PESQ | SI-SNRi |
| Unprocess | 0.7971 | 1.56 | - | 0.7992 | 1.62 | - | 0.7876 | 1.75 | - | 0.8032 | 1.88 | - |
| PHASEN | 0.8956 | 2.85 | 11.04 | 0.9142 | 2.94 | 11.46 | 0.8974 | 2.71 | 11.37 | 0.9076 | 2.90 | 11.69 |
| SN-Net | 0.9214 | 2.94 | 12.11 | 0.9362 | 3.02 | 11.98 | 0.9198 | 2.94 | 11.86 | 0.9298 | 3.00 | 12.06 |
| Inter-SubNet | 0.9226 | 2.92 | 11.96 | 0.9301 | 2.99 | 11.83 | 0.9213 | 2.99 | 12.01 | 0.9185 | 2.96 | 12.10 |
| CMGAN | 0.9354 | 3.03 | 12.36 | 0.9472 | 3.08 | 12.57 | 0.9387 | 3.08 | 12.34 | 0.9323 | 3.07 | 12.39 |
| MP-SENet | 0.9437 | 3.07 | 13.24 | 0.9521 | 3.13 | 13.32 | 0.9453 | 3.10 | 12.96 | 0.9453 | 3.14 | 13.50 |
| FIRING-Net | **0.9558** | **3.17** | **13.85** | **0.9643** | **3.21** | **14.12** | **0.9582** | **3.18** | **14.32** | **0.9541** | **3.23** | **14.16** |

## 4.2 MODEL SETTINGS AND EVALUATION METRICS

We trained the proposed model for 120 epochs with a batch size of 2, utilizing the Adam optimizer with an initial learning rate of 0.001. If the best model was not identified for 15 consecutive epochs, the learning rate was halved. Early stopping was employed, terminating the training if the best model was not found after 30 consecutive epochs. The STFT was applied using a Hanning window with a 32 ms window length and a 16 ms frame shift to convert the signal into the frequency domain. The number of channels **C** for all convolutional layers was set to 32. In the FRI-MHSA module, we used 8 attention heads. To mitigate overfitting, the dropout probability for all layers was set to 0.1. Four loss functions are utilized for model training: Magnitude Loss, Phase Loss, Complex Loss, and Time Loss. We use the following three commonly seen SE metrics for evaluation purposes. **PESQ:** wideband Perceptual Evaluation of Speech Quality (Rix et al., 2001). **STOI:** Short-Time Objective Intelligibility (Taal et al., 2011). **SI-SNRi:** Scale-Invariant Signal-to-Noise Ratio improvement (Le Roux et al., 2019). For all the metrics, the higher the score, the better the performance.

More details of experiential settings can be found in Appendix A.

## 5 RESULTS

### 5.1 MODEL COMPARISON

We conducted extensive experiments to quantitatively compare the SE performance of our proposed FIRING-Net with some existing SE models. Baseline models including PHASEN (Yin et al., 2020), SN-Net (Zheng et al., 2021), Inter-SubNet (Chen et al., 2023a), CMGAN (Abdulatif et al., 2024), and MP-SENet (Lu et al., 2023). PHASEN employs an interactive framework for jointly modeling magnitude and phase information, while SN-Net utilizes a dual-branch architecture to directly model the interaction between speech and noise features. Inter-SubNet explores cross-band dependencies through subband interaction mechanisms. In contrast, CMGAN and MP-SENet, despite being SOTA SE models, do not incorporate interactive feature modeling strategies. Note that these baseline models and the proposed FIRING-Net are trained on the same training set and evaluated on the same test set.

**WSJ0-SI 84+NOISEX-92:** This dataset provides a controlled environment combining speech with various noise types, making it ideal for evaluating the performance of SE models under standardized conditions. As detailed in Table 1, PHASEN, SN-Net, and Inter-SubNet fall short compared to CMGAN and MP-SENet due to less effective feature interaction strategies. PHASEN's interaction between magnitude and phase spectra is limited by the lack of distinct features in the phase spec-

Table 3: Ablative analysis of LM-E and LM-D by measuring STOI, PESQ, SI-SNRi, and number of trainable parameters.

| Encoder | Decoder | STOI | PESQ | SI-SNRi | Params. |
|---------|---------|------|------|---------|---------|
| LM-E | LM-D | 0.9135 | 3.03 | 13.96 | 1.81M |
| DCCM | DCCM | 0.8657 | 2.84 | 12.64 | 1.89M |
| LM-E | DCCM | **0.8854** | 2.93 | **12.89** | **1.87M** |
| DCCM | LM-D | 0.8782 | **2.94** | 12.71 | **1.87M** |
| FGCM | FGCM | 0.8738 | 2.88 | 12.84 | **1.78M** |
| LM-E | FGCM | 0.8927 | **2.95** | **13.27** | 1.80M |
| FGCM | LM-D | **0.9003** | 2.93 | 13.14 | 1.79M |

trum, making the interaction less meaningful. SN-Net attempts to model noise for interaction, but the unpredictable nature of noise undermines its effectiveness. Inter-SubNet's sub-band interaction focuses primarily on target speech modeling, offering limited improvement. In comparison, CMGAN and MP-SENet perform better with direct feature processing approaches, but FIRING-Net achieves superior results by introducing the Filter-Recycle-Interguide (FRI) framework. This method facilitates mutual guidance between target and non-target features, effectively refining speech information while suppressing noise. FIRING-Net achieves a PESQ score of 2.51 and STOI of 0.8414 under -5 dB SNR, significantly outperforming other models through more effective feature interaction.

**AVSpeech+AudioSet:** This dataset presents a more complex and varied set of challenges for speech enhancement models compared to the controlled WSJ0-SI 84 + NOISEX-92 environment. It includes a wide range of real-world noise types and speaker variations. As shown in Table 2, the complex and diverse noise types highlight the limitations of PHASEN, SN-Net, and Inter-SubNet. PHASEN's ineffective interaction between magnitude and phase spectra, SN-Net's reliance on noise prediction, and Inter-SubNet's focus on target speech modeling restrict their ability to generalize in dynamic noise conditions. FIRING-Net, leveraging its FRI framework, excels in handling overlapping and non-stationary noise. By using mutual guidance to refine speech and noise features, it outperforms CMGAN and MP-SENet across all metrics. For example, it achieves PESQ scores of 3.17 and 3.23 under babble and traffic noise, respectively, showcasing its robust generalization and effectiveness in diverse real-world scenarios.

## 5.2 ABLATION STUDY

We ablated the design choices and measured the average increase on the WSJ0-SI 84 + NOISEX-92 dataset. The following can be summarized from the ablation results:

**LM:** To assess the effectiveness of the proposed LM, we introduced two alternative modules: the densely connected convolutional module (DCCM), consisting of 6 cascaded convolutional layers (Pandey & Wang, 2020), and the fully gated convolutional module (FGCM), with 3 ELU-activated gated convolutional blocks (Tan & Wang, 2019). As shown in Table 3, both modules were configured to have a similar number of trainable parameters for a fair comparison. The most significant performance drop occurred when LM-E and LM-D were removed. FGCM, with its gating mechanism, effectively emphasized target features and outperformed DCCM. Overall, substituting LM-E or LM-D with DCCM or FGCM in the encoder or decoder led to a notable decline in performance across key metrics like STOI, PESQ, and SI-SNRi.

**GM:** To validate the effectiveness of the GM, we conducted two sets of comparative experiments, as presented in Table 4. The first set examines the FRI-MHSA module, where the $\mathbf{qk}^\top$ feature map is initially divided into four segments based on energy levels. We adjusted the number of segments and observed a significant performance drop when reduced to one segment (equivalent to standard MHSA). As the number of segments increases from 1 to 4, performance improves, demonstrating the advantage of subdividing the feature map and using high- and low-energy regions to guide intermediate feature extraction. However, performance plateaus beyond four segments, suggesting that further subdivision yields no additional benefit. Additionally, increasing the number of segments affects real-time processing efficiency. The second set of experiments assesses the GM's effectiveness in handling features across the time and frequency dimensions. Removing the processing of either dimension results in a significant performance decline.

Table 4: Ablation study on GM by measuring STOI, PESQ, SI-SNRi, number of trainable parameters, and real-time factor.

| Method | STOI | PESQ | SI-SNRi | Params. | CPU RTF |
|---|---|---|---|---|---|
| GM (*FRI-MHSA with 4 parts*) | 0.9135 | 3.03 | 13.96 | 1.81M | 0.60s |
| *-FRI-MHSA (1 Part)* | 0.8594 | 2.80 | 12.57 | **1.74M** | **0.39s** |
| *-FRI-MHSA (2 Parts)* | 0.8862 | 2.89 | 12.89 | 1.80M | 0.41s |
| *-FRI-MHSA (3 Parts)* | 0.8987 | 2.97 | 13.14 | 1.81M | 0.52s |
| *-FRI-MHSA (5 Parts)* | 0.9152 | 3.04 | 13.85 | 1.83M | 0.72s |
| *-FRI-MHSA (6 Parts)* | **0.9201** | **3.06** | **13.99** | 1.84M | 0.87s |
| w/o Frequency Processing | **0.8254** | 2.75 | 11.97 | 1.38M | 0.37s |
| w/o Time Processing | 0.8303 | **2.77** | **12.14** | 1.29M | 0.39s |
| w/o GM | 0.7631 | 2.64 | 10.52 | **1.03M** | **0.24s** |

Table 5: Ablation study on FRI Strategy in terms of STOI, PESQ, and SI-SNRi.

| Method | STOI | PESQ | SI-SNRi |
|---|---|---|---|
| GM | **0.9135** | **3.03** | **13.96** |
| *-Target Feature* | 0.8677 | 2.86 | 12.88 |
| *-Target and Original Features* | 0.8961 | 2.95 | 13.16 |
| *-Non-Target and Original Features* | 0.8306 | 2.69 | 9.52 |
| LM-E | **0.9135** | **3.03** | **13.96** |
| *-Target Feature* | 0.8738 | 2.84 | 12.83 |
| *-Target and Original Features* | 0.9003 | 2.92 | 13.05 |
| *-Non-Target and Original Features* | 0.7972 | 2.61 | 9.17 |
| LM-D | **0.9135** | **3.03** | **13.96** |
| *-Target Feature* | 0.8832 | 2.90 | 13.08 |
| *-Target and Original Features* | 0.9033 | 2.97 | **13.29** |
| *-Non-Target and Original Features* | 0.8992 | 2.93 | 13.38 |

**FRI Framework:** Table 5 demonstrates the effectiveness of this FRI strategy for SE. In the evaluation of LM, we observe that LM-E using non-target and original features performs worse than LM-E using target and original features. Additionally, LM-E *-Target Feature*, which processes only target features, highlights the importance of target features. LM-E also surpasses LM-E *-Target + Original Features*, confirming the value of non-target features and the effectiveness of the FRI framework in leveraging both feature types. For LM-D, results differ slightly. The performance of LM-D *-Target + Original Features* and LM-D *-Non-Target + Original Features* is similar and both exceed LM-D *-Target Feature*. This suggests that non-target information in the decoder contains useful speech components not involved in reconstruction, improving performance when both feature types are used. The evaluation of GM mirrors LM-E, reinforcing the effectiveness of the FRI framework. Notably, GM *-Target Feature* significantly outperforms GM *-FRI-MHSA (1 Part)* as shown in Table 4, indicating the superior efficacy of target feature extraction.

# 6 CONCLUSION

In this work, we propose an FRI framework that separates input features into target and non-target sets. These two sets guide each other to refine information, leading to the clustering of speech information in the target features and noise in the non-target features. We introduce FIRING-Net, a speech enhancement network comprising two main components: LM and GM. Both modules integrate interaction mechanisms to enable mutual guidance between the features, where speech is extracted from the non-target features and noise is filtered out from the target features. We conducted extensive experiments to validate the effectiveness of our method. From the results, we mainly conclude that: 1) Non-target features filtered by the SE network still contain speech information, and recycling them with the FRI strategy significantly boosts performance; 2) Mutual guidance between target and non-target features is crucial for filtering noise from the target and recovering speech from the non-target features. Future research focuses on designing lightweight model and enhancing real-time performance to enable deployment across various devices.

ACKNOWLEDGMENT

This work is supported by the National Natural Science Foundation of China (No.62471343, No. 62171326, No. 62071342).

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

## A    RECYCLING PROCESS

A pivotal aspect of the FRI strategy lies in recycling filtered information to derive non-target features. Inspired by Back-projection and Top-K techniques, we propose two distinct methodologies for non-target feature extraction: the calculation of residuals between input and output features and the utilization of energy distributions in the attention maps derived from self-attention mechanisms to identify and extract features corresponding to regions of low energy. These approaches are embedded within the LM and GM modules, facilitating a comprehensive and effective distinction between target and non-target features from both local and global analytical perspectives. To address the feasibility of the proposed methods for extracting non-target features, we emphasize their design based on direct calculations rather than relying on learnable modules. This ensures that the attributes of the extracted features align closely with our hypothesis, which predominantly associates non-target features with noise information.

**Residual Calculation Between Input and Output:** The extraction of non-target features using residuals is rooted in the concept of back-projection (Haris et al., 2018; 2019; Liu et al., 2019). This approach assumes that the information filtered out by the network represents the dominant noise components, making it highly relevant as non-target features. To ensure the success of this method, normalization layers are applied after each convolutional module. These layers maintain the energy range of the generated features within the scale of the input features, preventing the overall energy of the output from deviating. This step is critical, as unbounded variations in feature energy could distort the separation process and result in unintended feature overlaps or loss of relevant information. By maintaining consistency in energy distribution, the network effectively isolates noise-like elements as residuals, which then serve as refined non-target features.

**Utilization of Energy Distributions in Self-Attention Maps:** The second method employs the energy distributions in self-attention maps to extract non-target features. Specifically, attention mechanisms segment the feature space into regions of varying energy. High-energy regions correspond to speech-dominant components (target features), while low-energy regions predominantly contain noise-like characteristics. By calculating the cross-similarity between energy-segmented regions,

Table 6: Comparison with other methods on VoiceBank + DEMAND dataset. "-" denotes the result that is not provided in the original paper. Bold and underline indicate the best and second-best results.

| Models | Param. (M) | FLOPs (G) | WB-PESQ | CSIG | CBAK | COVL | STOI |
|---|---|---|---|---|---|---|---|
| Noisy | - | - | 1.97 | 3.35 | 2.44 | 2.63 | 0.91 |
| DEMCUS (Défossez et al., 2020) | 33.53 | 77.8 | 3.07 | 4.31 | 3.40 | 3.63 | 0.95 |
| TFT-Net (Tang et al., 2021) | 5.81 | 295.0 | 2.75 | 3.93 | 3.44 | 3.34 | - |
| SN-Net (Zheng et al., 2021) | - | - | 3.12 | 4.39 | 3.60 | 3.77 | - |
| DB-AIAT (Yu et al., 2022) | 2.81 | 68.0 | 3.31 | 4.61 | 3.75 | 3.96 | - |
| DPT-FSNet (Dang et al., 2022b) | **0.88** | **55.7** | 3.33 | 4.58 | 3.72 | 4.00 | 0.96 |
| CMGAN (Abdulatif et al., 2024) | 1.83 | 81.3 | 3.41 | 4.63 | 3.94 | 4.12 | 0.96 |
| TridentSE (Yin et al., 2023) | 3.03 | 59.8 | 3.47 | 4.70 | 3.81 | 4.10 | 0.96 |
| MP-SENet (Lu et al., 2023) | 2.05 | 84.7 | 3.50 | 4.73 | 3.95 | 4.22 | 0.96 |
| FIRING-Net(Proposed) | 1.81 | 64.2 | **3.57** | **4.79** | **3.98** | **4.33** | 0.96 |

this approach ensures a clean separation of target and non-target features without introducing cross-information. A critical design aspect here is the uniform application of energy segmentation and feature extraction across the same spectral representation. This prevents any inadvertent overlap between the extracted target and non-target features. Furthermore, the segmentation approach avoids the use of additional learnable parameters, ensuring that the extracted non-target features remain inherently noise-dominant, aligning with the hypothesis that noise information predominantly resides in low-energy regions.

## B  SUPPLEMENTARY EXPERIMENTAL SETUPS

### B.1  LOSS FUNCTION

We employ multi-level loss functions to train the proposed FIRING-Net. Following the approach in (Braun & Tashev, 2021; Lu et al., 2023; Abdulatif et al., 2024), we utilize time-domain loss ($\mathcal{L}_T$), magnitude loss ($\mathcal{L}_M$), and complex loss ($\mathcal{L}_C$), which are defined as:

$$\mathcal{L}_T = \mathbb{E}_{\mathbf{s},\hat{\mathbf{s}}}[||\mathbf{s} - \hat{\mathbf{s}}||_1], \qquad \mathcal{L}_M = \mathbb{E}_{\mathbf{S}_m,\hat{\mathbf{S}}_m}[||\mathbf{S}_m - \hat{\mathbf{S}}_m||_2^2], \tag{8}$$

$$\mathcal{L}_C = \mathbb{E}_{\mathbf{S}_r,\hat{\mathbf{S}}_r}[||\mathbf{S}_r - \hat{\mathbf{S}}_r||_2^2] + \mathbb{E}_{\mathbf{S}_i,\hat{\mathbf{S}}_i}[||\mathbf{S}_i - \hat{\mathbf{S}}_i||_2^2], \tag{9}$$

where $(\mathbf{S}_r, \mathbf{S}_i)$ and $(\hat{\mathbf{S}}_r, \hat{\mathbf{S}}_i)$ represent the real and imaginary parts of the clean and enhanced complex spectrogram. Previous works optimize the phase spectrogram within the complex spectrogram, since the absolute distance between two phases may not be their actual distance. Following the (Lu et al., 2023; Ai & Ling, 2023), we adopt anti-wrapping phase loss to optimize the phase spectrogram. The anti-wrapping phase loss includes three sub-losses, i.e., instantaneous phase loss $\mathcal{L}_{IP}$, group delay loss $\mathcal{L}_{GD}$, and instantaneous angular frequency loss $\mathcal{L}_{IAF}$, which are defined as:

$$\mathcal{L}_{IP} = \mathbb{E}_{\mathbf{S}_P,\hat{\mathbf{S}}_P}[||f_{AW}(\mathbf{S}_P - \hat{\mathbf{S}}_P)||_1], \tag{10}$$

$$\mathcal{L}_{GD} = \mathbb{E}_{\Delta_{DF}(\mathbf{S}_P,\hat{\mathbf{S}}_P)}[||f_{AW}(\Delta_{DF}(\mathbf{S}_P - \hat{\mathbf{S}}_P))||_1], \tag{11}$$

$$\mathcal{L}_{IAF} = \mathbb{E}_{\Delta_{DT}(\mathbf{S}_P,\hat{\mathbf{S}}_P)}[||f_{AW}(\Delta_{DT}(\mathbf{S}_P - \hat{\mathbf{S}}_P))||_1], \tag{12}$$

where $f_{AW}(t) = |t - 2\pi \cdot \text{round}(\frac{1}{2\pi})|, t \in \mathbb{R}$ is an anti-wrapping function, which is used to avoid the error expansion issue caused by phase wrapping. $\Delta_{DF}$ and $\Delta_{DT}$ represent the differential operators along the frequency axis and time axis, respectively. The anti-wrapping phase loss $\mathcal{L}_P$ is defined as:

$$\mathcal{L}_P = \mathcal{L}_{IP} + \mathcal{L}_{GD} + \mathcal{L}_{IAF}. \tag{13}$$

Finally, the total loss $\mathcal{L}_{Total}$ is the linear combination of $\mathcal{L}_T$, $\mathcal{L}_M$, $\mathcal{L}_C$, and $\mathcal{L}_P$,

$$\mathcal{L}_{Total} = \alpha_T \mathcal{L}_T + \alpha_M \mathcal{L}_M + \alpha_C \mathcal{L}_C + \alpha_P \mathcal{L}_P, \tag{14}$$

where $\alpha_T$, $\alpha_M$, $\alpha_C$, and $\alpha_P$ are hyperparameter and we follow (Lu et al., 2023) to set them to 0.2, 0.9, 0.1, and 0.3.

Table 7: Comparison with other methods on DNS Challenge Test Set. Bold and underline indicate the best and second-best results.

| Model | Feat. | Param.(M) | With Reverb | | | | Without Reverb | | | |
|---|---|---|---|---|---|---|---|---|---|---|
| | | | WB-PESQ | NB-PESQ | STOI | SI-SDR | WB-PESQ | NB-PESQ | STOI | SI-SDR |
| Noisy | - | - | 1.822 | 2.753 | 86.62 | 9.033 | 1.582 | 2.454 | 91.52 | 9.070 |
| DCCRN-E (Hu et al., 2020) | RI | 3.70 | - | 3.077 | - | - | - | 3.266 | - | - |
| Conv-TasNet (Luo & Mesgarani, 2019) | Waveform | 5.08 | 2.570 | - | - | - | 2.730 | - | - | - |
| PoCoNet (Isik et al., 2020) | RI | 50.00 | 2.832 | - | - | - | 2.748 | - | - | - |
| DCCRN+ (Lv et al., 2021) | RI | 3.30 | - | 3.300 | - | - | - | 3.330 | - | - |
| TRU-Net (Choi et al., 2021) | Mag | **0.38** | 2.740 | 3.350 | 91.29 | 14.87 | 2.860 | 3.360 | 96.32 | 17.55 |
| CTS-Net (Li et al., 2021) | Mag+RI | 4.99 | 3.020 | 3.470 | 92.70 | 15.58 | 2.940 | 3.420 | 96.66 | 17.99 |
| FullSubNet (Hao et al., 2021) | Mag | 5.64 | 3.057 | 3.584 | 92.11 | 16.04 | 2.882 | 3.428 | 96.32 | 17.30 |
| FullSubNet+ (Chen et al., 2022) | Mag+RI | 8.67 | 3.177 | 3.648 | 93.64 | 16.44 | 3.002 | 3.503 | 96.67 | 18.00 |
| TaylorSENet (Li et al., 2022) | RI | 5.40 | _3.330_ | _3.650_ | 93.99 | _17.10_ | 3.220 | 3.590 | 97.36 | 19.15 |
| SICRN (Zhao et al., 2024) | RI | 2.16 | 2.891 | 3.433 | 82.59 | 15.14 | 2.624 | 3.233 | 95.83 | 16.00 |
| CARNNHS (He et al., 2023) | DFT | - | 3.063 | 3.519 | 93.20 | 16.70 | 2.892 | 3.431 | 96.70 | 18.80 |
| MFNet (Liu et al., 2023) | RI | - | - | - | - | - | _3.430_ | _3.740_ | _97.78_ | **20.31** |
| FIRING-Net | Pha+Mag | _1.81_ | **3.632** | **3.863** | **96.65** | **17.93** | **3.505** | **3.781** | **98.09** | _20.10_ |

## B.2 EVALUATION METRICS

**PESQ**: Perceptual Evaluation of Speech Quality (PESQ) is a standard method for objectively assessing how speech quality is perceived by listeners (Rix et al., 2001). It provides an estimate of the subjective mean opinion score (MOS) for normal-hearing individuals, specifically evaluating audio quality in noisy or distorted telephone networks (Ma et al., 2009). The PESQ scale typically ranges from 1.0 to 4.5, making it a widely-used metric for measuring the performance of SE algorithms and the clarity of processed speech.

**STOI**: The Short-Term Objective Intelligibility (STOI) metric, which ranges from 0 to 1, provides an objective measure of speech intelligibility. It is particularly effective in evaluating speech in environments with temporally modulated noise or time-frequency processed signals, and is designed for normal-hearing listeners (Taal et al., 2011).

**SI-SNRi**: The Scale-Invariant Signal-to-Noise Ratio Improvement (SI-SNRi) is a metric used to assess the quality of enhanced speech. It is derived from the scale-invariant signal-to-noise ratio (SNR) (Le Roux et al., 2019), with higher values indicating better performance. SI-SNRi is defined as follows:

$$\text{SI-SNRi}(\mathbf{x}, \mathbf{s}, \hat{\mathbf{s}}) = \text{SI-SNR}(\mathbf{s}, \hat{\mathbf{s}}) - \text{SI-SNR}(\mathbf{s}, \mathbf{x}). \tag{15}$$

## C ADDITIONAL EXPERIMENTS

### C.1 EVALUATION ON BENCHMARK DATASET

To ensure a fair comparison between the proposed model and other SOTA models, we trained FIRING-Net on the training set of the public VoiceBank+DEMAND (Valentini-Botinhao et al., 2016) and DNS Challenge datasets and evaluated its performance on the test set.

**VoiceBank + DEMAND:** The VoiceBank + DEMAND test set includes two unseen speakers from the VoiceBank dataset (Veaux et al., 2013), with 20 distinct noise conditions. These conditions consist of five noise types from the DEMAND dataset (Thiemann et al., 2013b), each tested at four SNR levels (17.5, 12.5, 7.5, and 2.5 dB), resulting in a total of 824 test samples. Each test speaker has approximately 20 different sentences per condition. To evaluate and compare the enhanced speech quality across methods, we use mean opinion score (MOS) predictors: signal distortion (CSIG), background intrusiveness (CBAK), and overall quality (COVL), with scores ranging from 1 to 5 (Hu & Loizou, 2007). Additionally, wide-band PESQ (WP) and STOI are also employed. The averaged SE results on the VoiceBank + DEMAND dataset are presented in Table 6, in which we observe that the proposed FIRING-Net achieves superior performance than several SOTA SE models across all performance measures.

**DNS Challenge:** The proposed method is further evaluated on the DNS Challenge benchmark, comparing its performance against state-of-the-art (SOTA) approaches. Table 7 presents the average results across key metrics, including STOI, wide-band PESQ (WP), narrow-band PESQ (NP), and scale-invariant source-to-distortion ratio (SI-SDR) (dB). During training, noisy mixtures were generated with random SNRs ranging from -5 to 20 dB. As shown in Table 7, FIRING-Net consistently

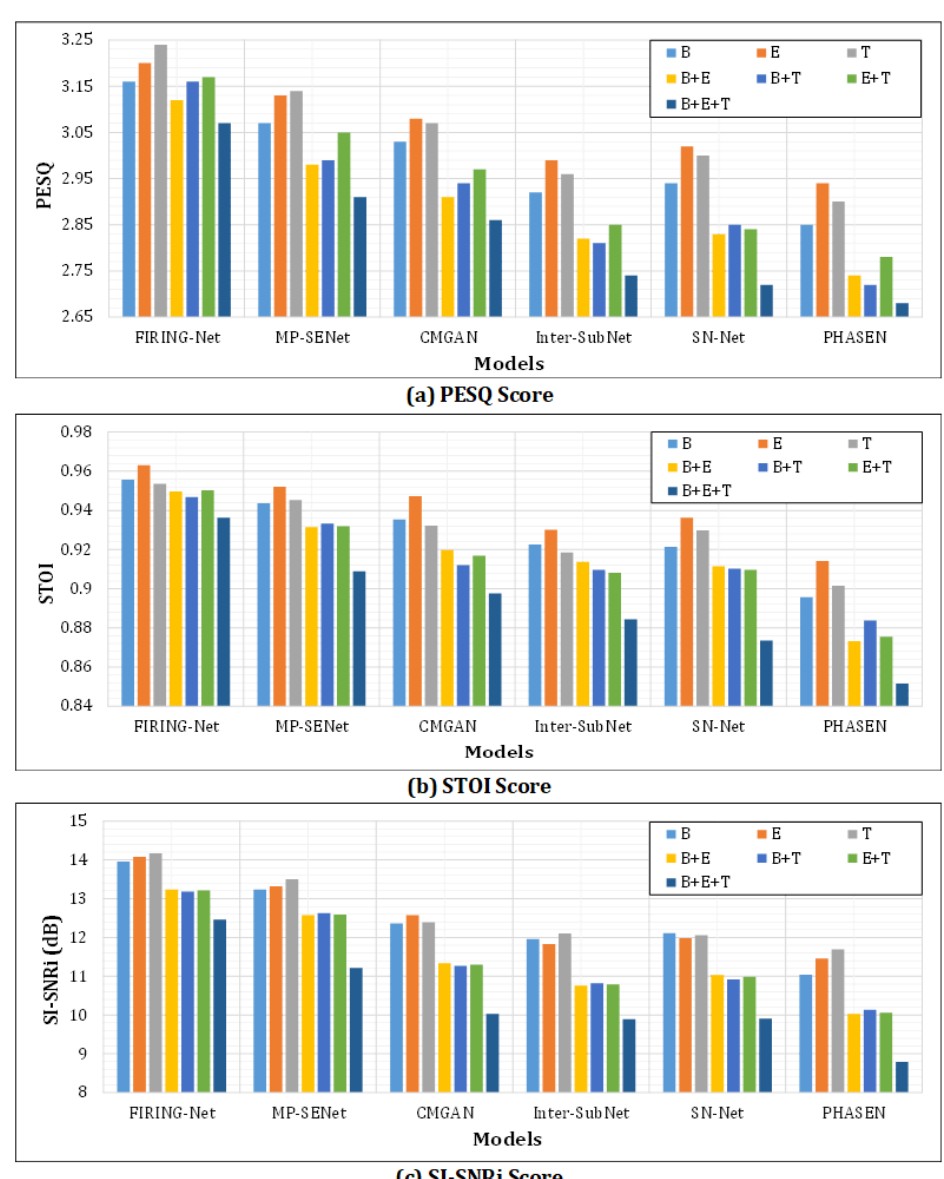

Figure 6: Comparison in PESQ, STOI, and SI-SNRi average for cases with multiple noise types, in which B, E, and T denote babble, engine, and traffic noisy types, respectively.

outperforms competing methods, demonstrating its robust speech enhancement capabilities on the DNS Challenge dataset.

## C.2 EVALUATION ON MORE COMPLEX NOISE CONDITIONS

We further show the advantages and flexibility of the proposed FIRING-Net by exploring FIRING-Net performance in environments where multiple types of noise are contained in the target domain (Lin et al., 2021), Figure 6 presents the PESQ and STOI results on multiple target noise types, where B, E, and T indicate "Babble", "Engine", and "Traffic", respectively. We observe that the results using 1, 2, and 3 noise types in the background environment are comparable. Figure 7 shows the enhanced spectrograms of selected baseline models and the proposed FIRING-Net. One can observe that the proposed model sufficiently preserves the spectral details while suppressing the residual noise over the selected baselines.

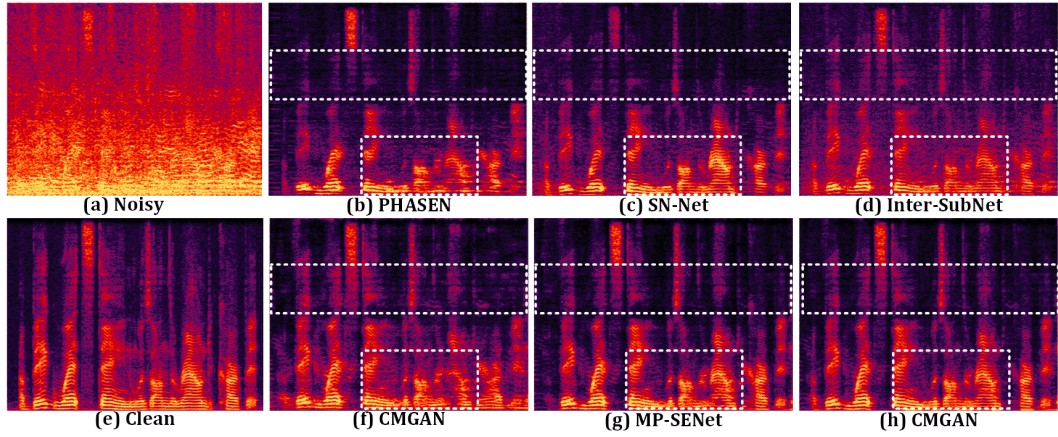

Figure 7: Visualization of spectrograms of enhanced speech generated from FIRING-Net and selected baseline models. The noisy sample contains two noise types, i.e., babble and engine, under -5dB SNR condition.

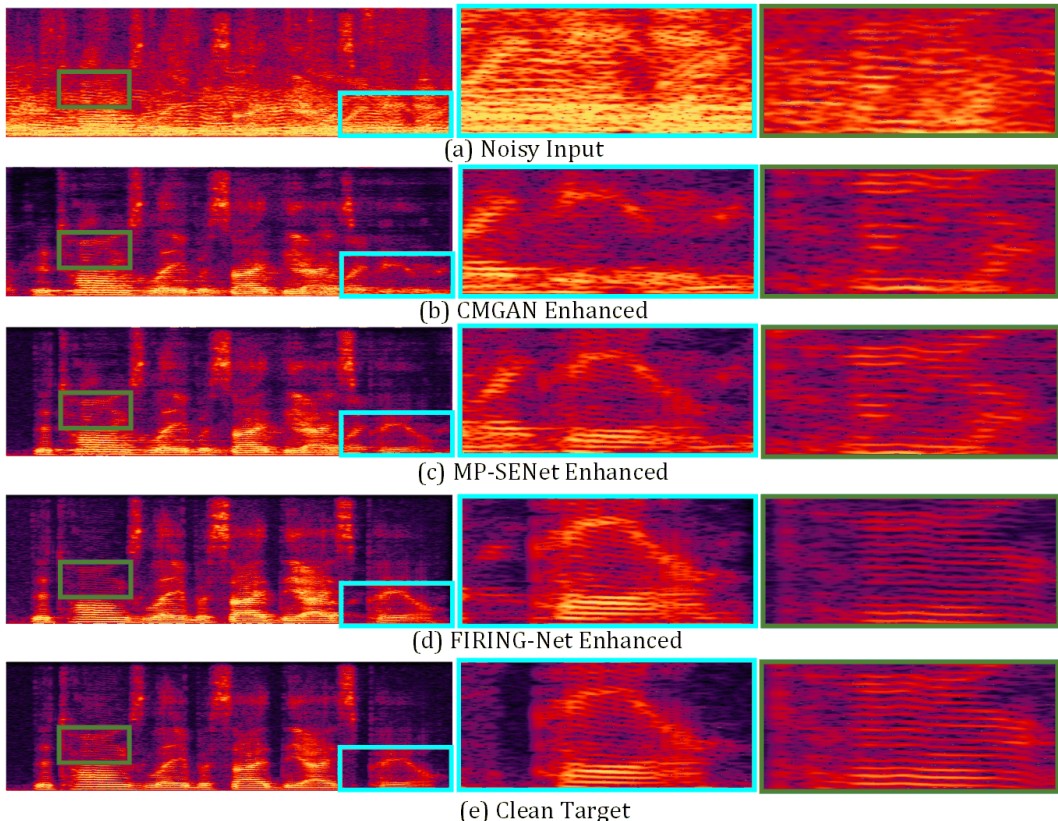

Figure 8: Visualization of spectrograms of enhanced speech generated from CMGAN, MP-SENet, and FIRING-Net in babble noise environment under -5dB SNR condition.

To better demonstrate the performance of FIRING-Net in scenarios where noise and speech components exhibit high similarity, we present spectrogram samples processed by CMGAN, MP-SENet, and FIRING-Net for Babble noise and Factory noise. As shown in Figure 8, Babble noise consists

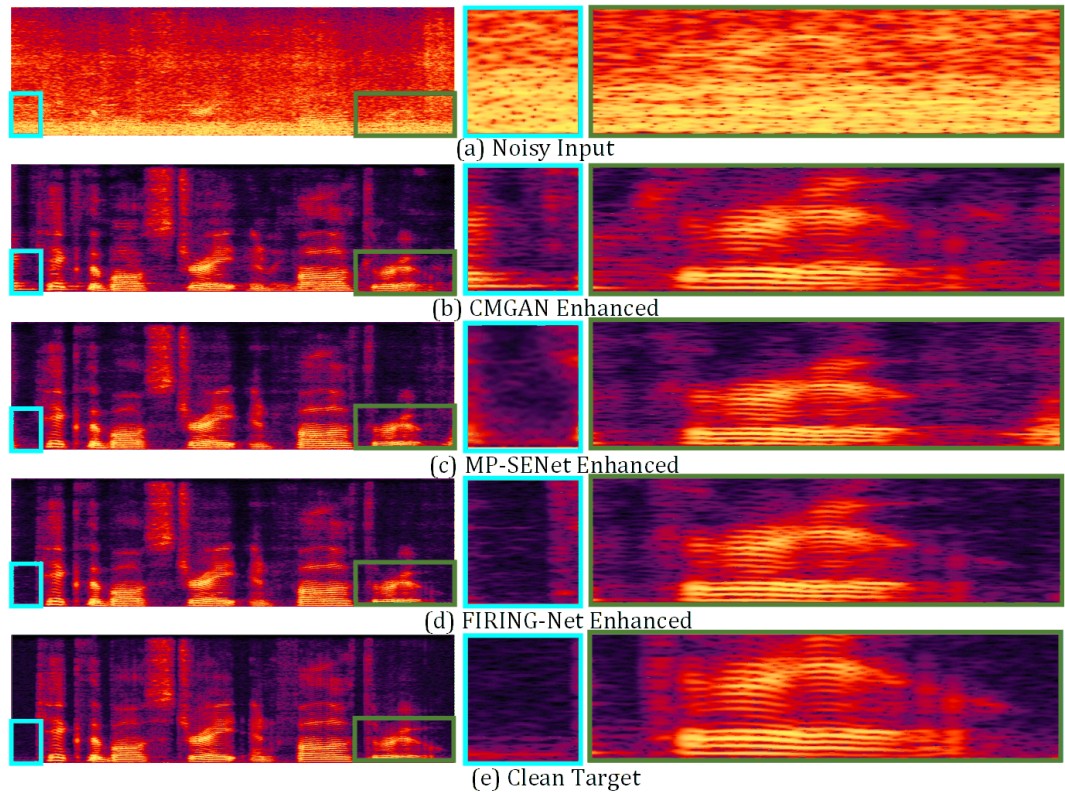

Figure 9: Visualization of spectrograms of enhanced speech generated from CMGAN, MP-SENet, and FIRING-Net in factory noise environment under -5dB SNR condition.

of background noise heavily influenced by speech interference from multiple speakers, making its acoustic characteristics more similar to the target speech compared to other noise types. Similarly, as depicted in Figure 9, Factory noise was selected due to its composition of low-SNR white noise, which heavily masks speech signals. This masking effect suppresses the inherent acoustic characteristics of the speech, making it challenging to distinguish from the background noise. Figure 8 and Figure 9 illustrate the spectrograms under Babble and Factory noise conditions, respectively. The two rightmost columns provide a detailed view of the corresponding spectrograms on the left. From these figures, it is evident that in such extreme cases, where noise and speech components are highly similar and difficult to differentiate, CMGAN and MP-SENet struggle to effectively extract and restore the speech components. In contrast, FIRING-Net leverages the Filter-Recycle-Interguide (FRI) strategy to achieve a more complete restoration of speech components in these challenging scenarios.

Additionally, to evaluate the robustness of FIRING-Net under extremely challenging noise environments with very low signal-to-noise ratios, we conducted comparative experiments against baseline models on the WSJ0-SI 84+NOISEX-92 test set under -10 dB SNR conditions, specifically in Babble and Factory noise environments. The results, presented in Table 8, demonstrate that FIRING-Net consistently outperforms the baseline models, achieving superior performance across all evaluated metrics. This validates its capability to effectively handle diverse and complex noise scenarios, particularly under extreme SNR conditions where traditional approaches often struggle.

## C.3 EVALUATION ON REAL-WORLD SCENARIOS

The performance of FIRING-Net and the selected baseline models was initially evaluated on the AVSpeech + AudioSet dataset, a diverse corpus featuring a wide range of speakers and noise types. The results highlighted FIRING-Net's superior performance, suggesting its strong potential for real-

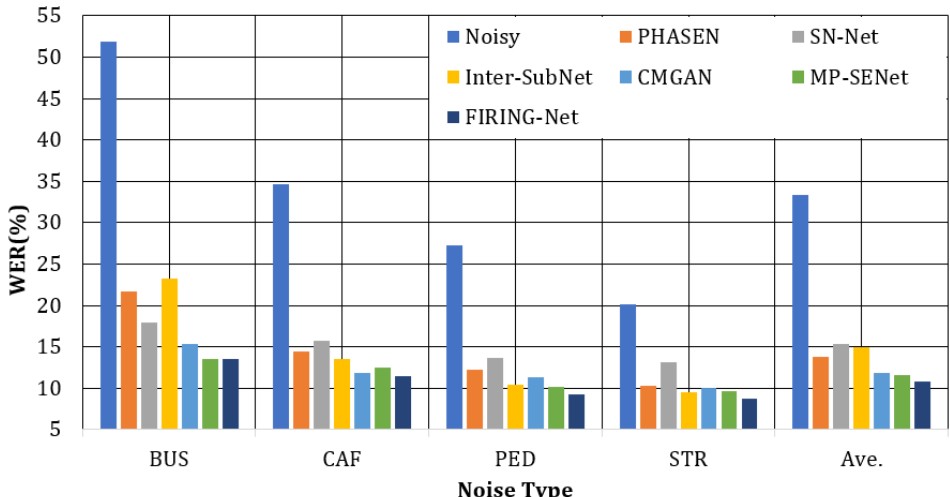

Figure 10: ASR performance on FIRING-Net and baseline-enhanced speech samples on CHiME-3 real data test set using DNN based ASR system.

Table 8: Comparison with selected baseline models on WSJ0-SI 84+NOISEX-92 under -10 dB SNR level in "Babble" and "Factory" noise environments. Bold and underline indicate the best and second-best results.

| Model | Factory | | | Babble | | |
|---|---|---|---|---|---|---|
| | STOI | PESQ | SI-SNRi | STOI | PESQ | SI-SNRi |
| Noisy | 0.4452 | 1.15 | - | 0.4338 | 1.07 | - |
| PHASEN | 0.5731 | 1.58 | 11.32 | 0.4882 | 1.39 | 8.58 |
| SN-Net | 0.6352 | 1.63 | 11.65 | 0.5011 | 1.45 | 9.21 |
| Inter-SubNet | 0.6058 | 1.60 | 11.53 | 0.4945 | 1.46 | 9.14 |
| CMGAN | 0.6627 | 1.66 | 11.87 | 0.5259 | 1.51 | 9.86 |
| MP-SENet | 0.6584 | 1.68 | 12.24 | 0.5827 | 1.53 | 10.21 |
| FIRING-Net | 0.7286 | 1.83 | 13.36 | 0.7156 | 1.77 | 12.38 |

world applications. To further validate its effectiveness under practical conditions, we conducted experiments using the CHiME-3 real-world test set (Barker et al., 2017), which includes noisy speech recorded in real-life environments such as public transport (BUS), cafeteria (CAF), street junction (STR), and pedestrian area (PED). For evaluation, we employed the official DNN-based ASR model provided by CHiME-3 (Barker et al., 2017), comprising seven layers with 2048 units per hidden layer. The input layer incorporated 5 frames of left and right context (i.e., 11×40=440 units). The model was trained using a standard pipeline, including pre-training with restricted Boltzmann machines, cross-entropy training, and sequence-discriminative training using the state-level minimum Bayes risk (sMBR) criterion (Burget et al., 2013). Using this ASR system, we measured the Word Error Rate (WER) after speech enhancement, with results presented in Figure 10. FIRING-Net consistently achieved the lowest WER across all tested environments, surpassing the baseline models. This demonstrates its capability to manage the complexities of real-world noise while preserving speech intelligibility. The outstanding performance can be attributed to the FRI framework, which facilitates effective interaction between target and non-target features, enabling precise noise suppression and robust speech enhancement. These findings underscore FIRING-Net's robustness and efficacy in handling diverse and dynamic noise conditions in practical applications.

## C.4 EVALUATION ON IM-E AND IM-D

As discussed earlier, the design concepts of IM-E and IM-D are distinct. The design of IM-E focuses on the mutual guidance between target and non-target features, aiming to extract speech information with high similarity to the target features from the non-target features and supplement it into the target features. At the same time, noise information that is highly similar to the non-target features is filtered out from the target features, as the features in the encoder often contain a substantial

Table 9: Ablation study on IM-E and IM-D by replacing them with other modules. Bold and underline indicate the best and second-best results.

| LM-E | GM | LM-D | STOI | PESQ | SI-SNRi | Param.(M) |
|------|------|------|------|------|---------|-----------|
| IM-E | IM-E | IM-D | **0.9135** | **3.03** | **13.96** | 1.81 |
| IM-E | IM-E | IM-E | 0.9103 | 3.01 | 13.85 | 1.72 |
| IM-D | IM-D | IM-D | 0.8762 | 2.96 | 12.87 | **1.62** |
| ConvBlock | ConvBlock | ConvBlock | 0.8226 | 2.77 | 11.63 | 1.89 |
| ConvBlock | ConvBlock | IM-D | 0.8307 | 2.76 | 11.72 | 2.03 |
| IM-E | IM-E | ConvBlock | 0.8812 | 2.98 | 12.94 | 1.82 |

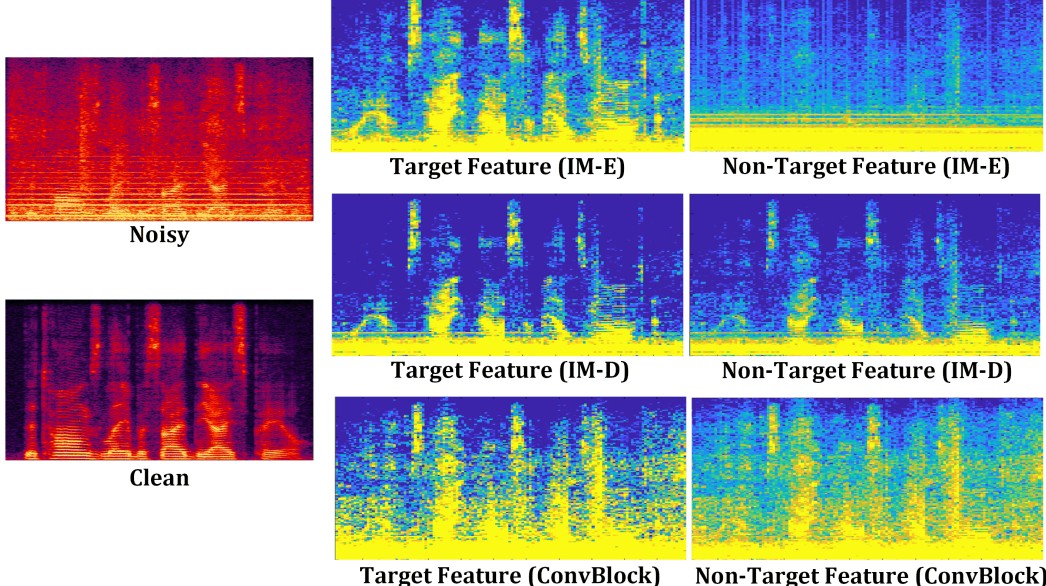

Figure 11: Visualization of feature maps captured from the first LM-E block and the LM-E when IM-E is replaced by IM-D and ConvBlock.

amount of noise. In contrast, the design of IM-D is based on the observation that the features in the decoder typically contain little to no noise. In this stage, the non-target features usually consist of speech information that has not been involved in the reconstruction process. Therefore, the design of IM-D primarily considers how to better fuse the speech information from both target and non-target features through mutual guidance, ensuring their effective integration in the subsequent speech reconstruction process. Therefore, we validate the effectiveness of IM-E and IM-D by replacing these modules and evaluating the performance of their internal structures when disassembled.

**Replacing IM-E and IM-D**: For fairness in comparison, we replace IM-E and IM-D with a configuration consisting of four convolutional blocks (each block containing a convolutional layer, batch normalization, and a PReLU activation function) to avoid performance differences arising from variations in the number of trainable parameters. The evaluation results are shown in Table 9. Firstly, we replaced all IM-D modules in LM-D with IM-E, which resulted in a slight performance decrease for FIRING-Net, along with a noticeable increase in model complexity. Next, we replaced all IMs in the network with IM-D and ConvBlock. This led to a significant drop in model performance, with ConvBlock performing worse than IM-D. To investigate the cause, we compared the relevant and irrelevant feature maps of LM-E using IM-E, IM-D, and ConvBlock, as shown in Figure 8. The figure reveals that with IM-E, speech and noise information are distinctly aggregated into the target and non-target features, respectively. In contrast, with IM-D and ConvBlock, the separation of speech and noise information within target and non-target features is not as clear, and ConvBlock exhibits more noise in both target and non-target features compared to IM-D. This suggests that the ability of LM-E to extract speech information into target features and noise information into non-target

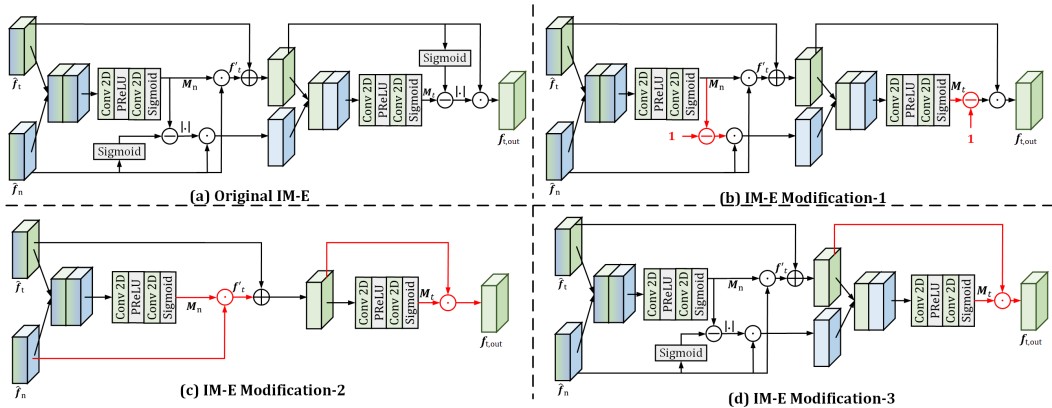

Figure 12: Network architecture of three modifications of IM-E. (a) We replace the "$|Sigmoid(\hat{\mathbf{f}}_n) - \mathbf{M}_n|$" and "$|Sigmoid(\hat{\mathbf{f}}_t) - \mathbf{M}_t|$" with "$1 - \mathbf{M}_n$" and "$1 - \mathbf{M}_t$". (b) We remove the non-target feature to guide the target feature for noise information extraction. (c) We remove the "$|Sigmoid(\hat{\mathbf{f}}_t) - \mathbf{M}_t|$". These modifications are marked in red.

Table 10: Evaluation on the structure of IM-E. Bold and underline indicate the best and second-best results.

| Module | STOI | PESQ | SI-SNRi | Param.(M) |
|---|---|---|---|---|
| IM-E | **0.9135** | **3.03** | **13.96** | 1.81 |
| IM-E Modification-1 | 0.8884 | 2.94 | 13.25 | 1.81 |
| IM-E Modification-2 | 0.9008 | 2.99 | 13.51 | **1.59** |
| IM-E Modification-3 | 0.8987 | 2.96 | 13.67 | 1.81 |

features is closely related to the structure of the interaction modules, highlighting the effectiveness of IM-E.

**Internal structure disassembling for IM-E**: IM-E incorporates three unique structures. First, it reverses the features emphasized in $\mathbf{M}_n$ and $\mathbf{M}_t$ (for example, $\mathbf{M}_n$ emphasizes speech information in $\mathbf{f}_n$, and after reversal, we want it to emphasize noise information). Instead of subtracting these masks directly from 1, we subtract them from $|Sigmoid(\hat{\mathbf{f}}_n)|$ and $|Sigmoid(\hat{\mathbf{f}}_t)|$ respectively, and then take the absolute value (This modified structure is shown in Figure 12(a)). Second, it uses $\mathbf{M}_n$, which is the complement of the one used to extract speech information from non-target features, to instead extract noise information. This noise information is then employed to guide the extraction of noise from the target features, i.e., $|Sigmoid(\hat{\mathbf{f}}_n) - \mathbf{M}_n|$ (This modified structure is shown in Figure 12(b)). Third, during noise filtering of the target feature, another mask $\mathbf{M}_t$, is reversed; initially used for filtering noise and extracting speech, it is reversed to instead extract noise, i.e., $|Sigmoid(\hat{\mathbf{f}}_t) - \mathbf{M}_t|$ (This modified structure is shown in Figure 12(c)). The results are presented in Table 10. First, we observed that the performance of IM-E Modification-1 is significantly lower than that of IM-E. Both structures attempt to reverse the features emphasized by the sigmoid activation function. To explain this phenomenon, we present the feature spectrograms of $|Sigmoid(\hat{\mathbf{f}}_n) - \mathbf{M}_n|$ and $1 - \mathbf{M}_n$ in Figure 15. From this, we can see that the $1 - \mathbf{M}_n$ operation amplifies all features that are not emphasized by $\mathbf{M}_n$, and this amplification primarily boosts regions in the feature spectrum with originally weaker energy, introducing new interfering features into the feature processing. Additionally, we observe a performance drop in IM-E after modifications 2 and 3, with IM-E Modification-3 performing worse than Modification-2. This is because, in both modifications, $\mathbf{M}_t$ aggregates speech information. However, in Modification-3, $\mathbf{M}_t$ is generated from fused target and non-target features, allowing noise from the non-target features to interfere with the precision of $\mathbf{M}_t$. In the original IM-E, the operation $|Sigmoid(\hat{\mathbf{f}}_t) - \mathbf{M}_t|$ shifts the mask's function from aggregating speech to aggregating noise, thus enhancing the non-target features' contribution.

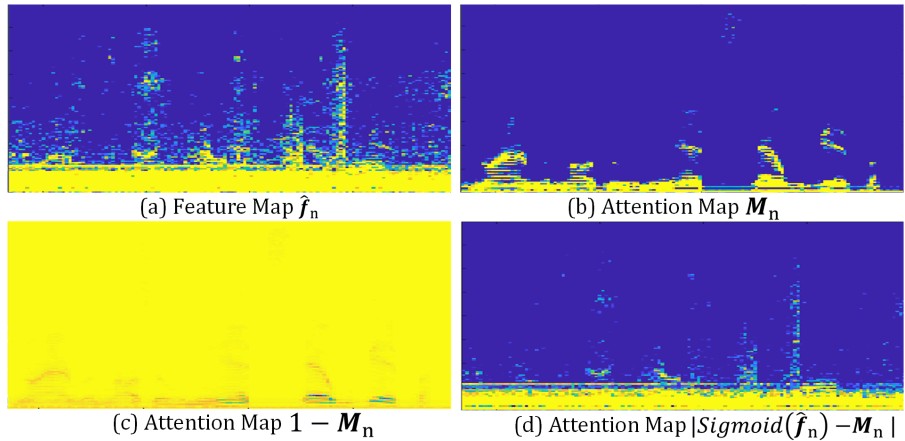

(a) Feature Map $\hat{f}_n$        (b) Attention Map $\boldsymbol{M}_n$

(c) Attention Map $1 - \boldsymbol{M}_n$        (d) Attention Map $|Sigmoid(\hat{\boldsymbol{f}}_n) - \boldsymbol{M}_n|$

Figure 13: Visualizations of feature and attention maps of $\hat{\mathbf{f}}_n$, $\mathbf{M}_n$, $1 - \mathbf{M}_n$, and $|Sigmoid(\hat{\mathbf{f}}_n) - \mathbf{M}_n|$.

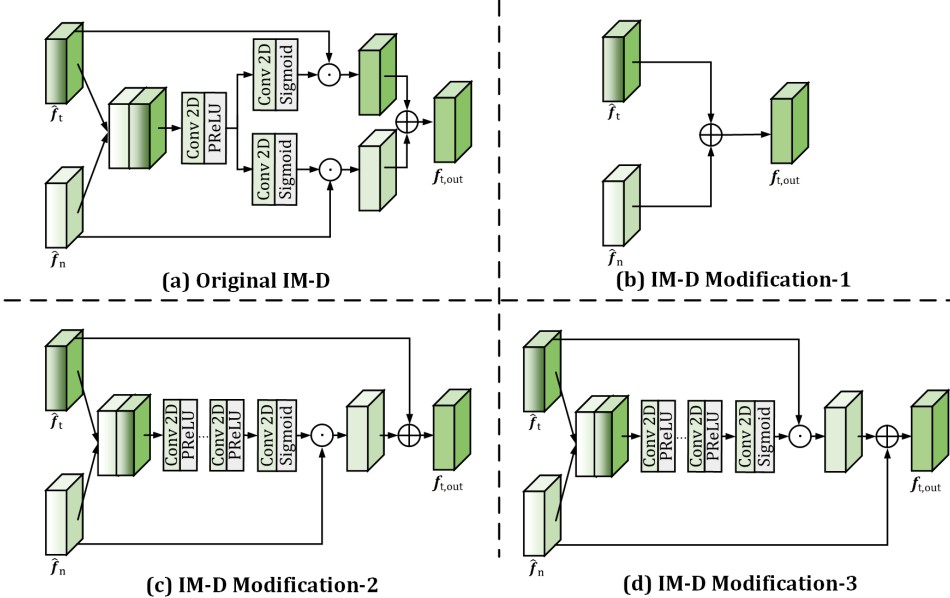

Figure 14: Network architecture of three modifications of IM-D. (a) We directly add the target and non-target features. (b) We process the non-target feature only to add with target feature. (c) We process the target feature only to add with non-target feature.

**Internal structure disassembling for IM-D**: The structure of IM-D can be roughly summarized as a weighted summation of target and non-target features. To validate the effectiveness of this design, we made three modifications to IM-D, as shown in Figure 10. In the first modification, we directly sum the target and non-target features without applying any weights. In the second modification,

Table 11: Evaluation on the structure of IM-D. Bold and underline indicate the best and second-best results.

| Module | STOI | PESQ | SI-SNRi | Param.(M) |
|---|---|---|---|---|
| IM-D | **0.9135** | **3.03** | **13.96** | 1.81 |
| IM-D Modification-1 | 0.8762 | 2.95 | 13.23 | 1.23 |
| IM-D Modification-2 | 0.8987 | 2.98 | 13.82 | 1.81 |
| IM-D Modification-3 | 0.8934 | 2.97 | 13.76 | 1.81 |

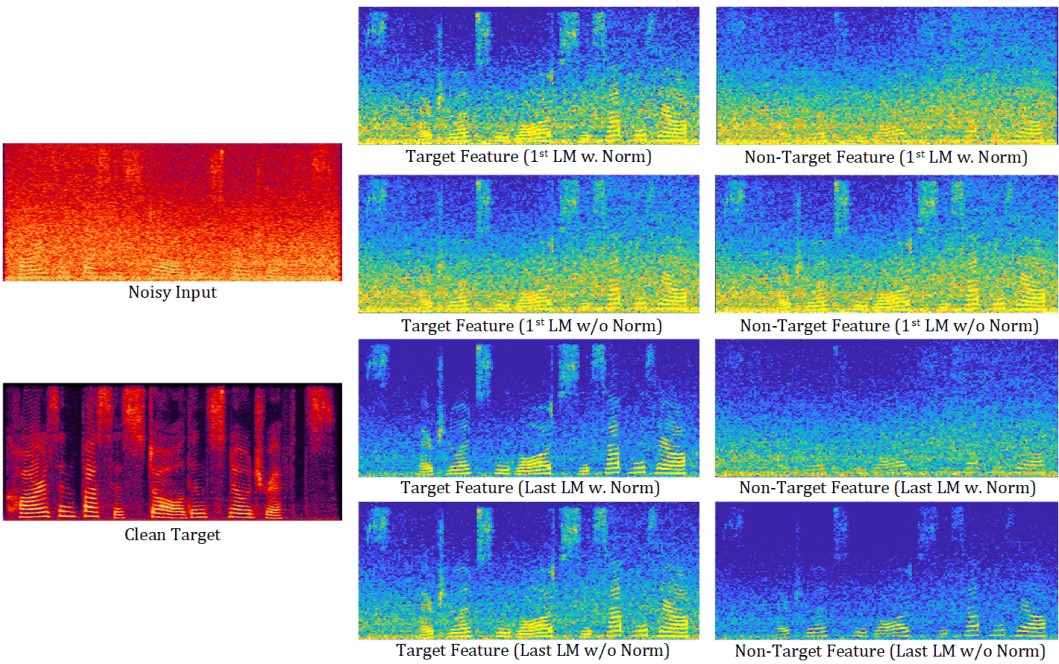

Figure 15: Visualizations of target and non-target features extracted by LM-E with and without normalization operation, along with their corresponding spectrograms of noisy mixtures and clean target speech.

Table 12: Evaluation of the effect of normalization and the choice between addition and subtraction operations for the LM module.

| Method | STOI | PESQ | SI-SNRi |
|---|---|---|---|
| LM-E | 0.9135 | 3.03 | 13.96 |
| -w/o *Norm + Subtraction* | 0.8338 | 2.76 | 12.25 |
| -w. *Norm + Addition* | 0.8820 | 2.87 | 13.11 |
| -w/o *Norm + Addition* | 0.8454 | 2.78 | 12.18 |
| LM-D | 0.9135 | 3.03 | 13.96 |
| -w/o *Norm + Subtraction* | 0.8557 | 2.81 | 12.54 |
| -w. *Norm + Addition* | 0.8739 | 2.85 | 12.89 |
| -w/o *Norm + Addition* | 0.8420 | 2.82 | 12.73 |

only the non-target features are weighted, while in the third modification, only the target features are weighted. To ensure a fair comparison, we increased the number of convolutional blocks in the second and third modifications so that the number of trainable parameters matches that of the original IM-D. The results, as shown in Table 11, indicate that the performance of IM-D Modification-1, where target and non-target features are summed without weighting, is noticeably worse compared to IM-D Modification-2 and IM-D Modification-3, where only one feature is weighted. Additionally, while the performance difference between IM-D Modification-2 and IM-D Modification-3 is not significant, both modifications still perform significantly worse than the original IM-D.

## C.5 ANALYSIS OF RESIDUAL CALCULATION IN LM

In the LM, residual computation is employed to extract the filtered non-target information, with the objective of maximizing the inclusion of noise-related features. However, the convolutional operations within the LM can cause a mismatch in the overall energy range between the input and output features. This discrepancy compromises the ability of the residuals to effectively isolate noise-dominant features, as illustrated in Figure 15. To address this issue, normalization is applied after each convolutional layer in the LM, ensuring that the input and output features maintain comparable

Table 13: Evaluation on Phase Decoder. Bold and underline indicate the best and second-best results.

| Phase Decoder | STOI | PESQ | SI-SNRi |
|---|---|---|---|
| LM-D | **0.9135** | **3.03** | **13.96** |
| DCCM | 0.9074 | 2.98 | 13.10 |
| FGCM | 0.9103 | 2.99 | 13.54 |
| Noisy Phase | 0.8621 | 2.89 | 12.62 |
| Clean Phase | 0.9296 | 3.09 | 14.83 |

energy levels. This adjustment effectively mitigates the aforementioned challenge. Additionally, we present the results in Table 12 to evaluate the impact of removing the normalization operation on the subtraction process, as well as the performance effects of replacing the subtraction operation with addition, both with and without normalization. The comparative results indicate that when the normalization operation is removed, i.e., the features are unconstrained, there is little performance difference between subtraction and addition. However, when normalization is applied, constraining the features, the performance of the subtraction operation is significantly superior to that of the addition operation.

## C.6    EVALUATION ON PHASE DECODER

Since phase information, unlike magnitude information, lacks a clear structural pattern, we conducted an evaluation to validate the effectiveness of the proposed FIRING-Net's Phase Decoder. For comparison, we replaced LM-D with DCCM and FGCM, and compared the performance of the phase output from the Phase Decoder with both the noisy input phase and the clean target phase. As shown in Table 13, although the performance of the speech synthesized with the Phase Decoder-generated phase is weaker than that of the clean phase, it significantly outperforms both the noisy phase and the phase produced by DCCM and FGCM-based structures. However, it is important to note that the mapping relationship between noisy and clean phase is less explicit compared to that of noisy and clean magnitude. Thus, while our proposed FRI framework theoretically does not fully apply to phase recovery, the use of IM-D in the Phase Decoder yields performance gains. Whether these gains align with the theoretical improvements described by the FRI framework remains uncertain and will be explored in future work, with a focus on developing models better suited for phase recovery.

