# OpenReview forum: "FIRING-Net: A filtered feature recycling network for speech enhancement"
_ICLR.cc/2025/Conference — ICLR 2025 Poster_

### Official Review · Reviewer_9ACG · 2024-10-31

**Soundness:** 3
**Presentation:** 3
**Contribution:** 3
**Rating:** 6
**Confidence:** 4

**Summary:**

This paper, based on the challenges in the field of speech enhancement when noise and target speech are similar, designs a Filter-Recycle-InterGuide Network (FRIGNet). It utilizes target features and non-target features for mutual refinement, and based on the mutual refinement of positive and negative features, it has designed LM and GM modules. Additionally, it employs an attention module based on energy segmentation to complement the network.

**Strengths:**

The paper has a novel approach. While others focus only on how to better eliminate non-target features, the authors consider both target features and non-target features to refine each characteristic. Based on this, the overall network architecture and individual modules are designed based on the logic of mutual refinement between target features and non-target features. An attention mechanism based on energy levels is also introduced. The model performs well in both comparative experiments and ablation studies.

**Weaknesses:**

The paper extensively discusses how the model uses operations such as generating masks, convolutions, and additions/subtractions to obtain the so-called target features and non-target features mentioned in the text. However, it lacks a theoretical explanation for why these operations result in target features or non-target features. Additionally, it is unclear whether the attention module based on energy segmentation is an original innovation or borrowed from other papers.

**Questions:**

N/A

---

> ### Author Response · Authors · 2024-11-23
> **Response to Reviewer 9ACG**
>
> Thank you for your insightful comments. We appreciate the opportunity to clarify the theoretical foundation behind how the model extracts target and non-target features, as well as the originality of the attention module based on energy segmentation.
>
> Q1: The paper extensively discusses how the model uses operations such as generating masks, convolutions, and additions/subtractions to obtain the so-called target features and non-target features mentioned in the text. However, it lacks a theoretical explanation for why these operations result in target features or non-target features. Additionally, it is unclear whether the attention module based on energy segmentation is an original innovation or borrowed from other papers.
>
> A1:The key operations for obtaining target and non-target features in our model involve techniques such as residual calculation, energy segmentation, and the use of interaction modules. Here, we provide a more detailed theoretical explanation for why these operations lead to the separation of target and non-target features:
>
> 1. Residual Calculation (for LM):
>
> The extraction of non-target features relies on the concept of residuals, which is grounded in the back-projection principle. The core assumption is that the information filtered out by the network primarily corresponds to noise. By calculating the residuals between the input and output features, we isolate components that deviate from the target speech features, which are dominated by noise. The role of normalization layers after each convolution is to maintain the energy consistency between the input and the extracted residuals, ensuring that the non-target features are not distorted and are effectively separated as noise-like residuals. This operation is crucial for distinguishing non-target features from the desired target features.
>
> 2. Energy Distributions in Self-Attention Maps (for GM):
>
> The second method uses energy distributions in self-attention maps to segment the feature space. This approach assumes that target features, which are speech-dominant, reside in high-energy regions of the feature map, while non-target features, primarily composed of noise, are located in low-energy regions. The energy-based segmentation allows the model to focus on regions of the feature space that contain noise-like elements, ensuring clean separation of target and non-target features. The uniform application of this segmentation approach across the same spectral representation prevents overlap between features and avoids the introduction of learnable parameters, preserving the assumption that non-target features are inherently noise-dominant.
>
> Regarding your inquiry about the originality of the attention module based on energy segmentation, we would like to clarify that while the concept of energy-based segmentation shares some commonality with previous approaches [1], the specific implementation in our model is novel. The use of energy distributions in self-attention maps for segmenting feature space into regions of high and low energy is an adaptation tailored to our goal of separating target (speech) and non-target (noise) features. While energy-based segmentation techniques have been explored in other contexts, our approach integrates this concept within the framework of self-attention mechanisms, specifically designed to address the challenges of speech separation in noisy environments. This integration ensures an efficient separation of speech and noise without relying on additional learnable parameters, which distinguishes our method from prior work.
>
> We have included a dedicated section in the revised Appendix A that analyzes the feasibility of the Recycling Process, offering a clearer theoretical explanation of how these operations contribute to the effective distinction between target and non-target features.
>
> [1] Chen X, Li H, Li M, Pan J. Learning a sparse transformer network for effective image deraining. InProceedings of the IEEE/CVF Conference on Computer Vision and Pattern Recognition 2023 (pp. 5896-5905).

---

### Official Review · Reviewer_S2co · 2024-10-31

**Soundness:** 2
**Presentation:** 2
**Contribution:** 2
**Rating:** 6
**Confidence:** 3

**Summary:**

This paper presents a new deep network architecture for speech enhancement (SE) called FIRING-Net (FIlter-Recycle-INterGuide NETwork). To deal with the more challenging scenarios where the interference/noise characteristics are highly similar to the target speech signal, e.g., in the case of babble noise, the authors propose to recycle filtered-out features for extracting more discriminative patterns between speech and noise through refinement. This is achieved by re-defining the filtered-out features as non-target features, which are used to guide the learning of target features in an interactive manner. More specifically, the FIRING-Net features two key components: Local Module (LM) and Global Module (GM), to perform the recycle-and-refine idea in a local and global manner. The LMs utilizes Interaction Modules (IMs) that have different designs for the encoder and decoder sides. The GM exploits energy distributions of the attention maps to guide extration of the speech and noise features. Experiments on several datasets demonstrate the effectiveness of FIRING-Net over various signal-to-noise ration (SNR) settings and across several noise types, including the challenging case of babble noise.

**Strengths:**

- The proposed system (FIRING-Net) based on the idea of recycling intermediate speech and noise features to further improve the separation of target and non-target components seems novel.

- Good performance of the FIRING-Net across various SNR and noise types (E.g., Tables 1, 2).

- Relatively small model size with low latency (Eg., Tables 3, 4), which is important for many SE applications.

- Extensive ablation studies showing the effect and contribution of each key component, i.e. LM, GM and IMs, on the overall SE performance.

**Weaknesses:**

- The main weakness is there seems lacking a clear explanation as why the proposed architecture, especially the Local Modules (LMs), can promote target speech and non-target noise features extraction in the way as claimed in the paper. To be more specific, for the proposed LM-E (and LM-D) in Figure 2, what is the actual mechanism or reason that leads to the model assigning speech to the $f_t$ branch and noise to the $f_n$ branch? Please provide more detailed explanations on how such network design promotes speech and noise separation  to carry out the proposed feature recycling scheme.

- Another weakness of the paper is that the proposed SE model is tested on synthesized noisy speech data only. To better demonstrate the superiority of the proposed model, it will be nice if testing on real-world recorded noisy speech samples can be performed (e.g., using the realistic noisy samples from the CHiME-3 dataset: *J. Barker et al., “The third ‘CHiME’ speech separation and recognition challenge: Analysis and outcomes,” Comput. Speech Lang, 2017*). In this case, since there will be no ground-truth references to compute the objective metric scores used in the paper (i.e., PESQ, STOI, and SI-SNRi), the comparison with baselines may be done by using subjective listening testing or using automatic speech recognition (ASR) as evaluators.

**Questions:**

- Could the authors explicitly associate the "Network for speech extraction" in Figure 1 to the corresponding module(s) or layer(s) in Figure 2? This will help understanding where the recycling scheme actually takes place. Similarly, association of the "Network for noise extraction" in Figure 1 to the place(s) in Figure 2 will be helpful.

- In Figure 2, the LM-E (or LM-D) module has addition and subtraction operators. However, the subtraction operation may not lead to a different result from using addition if the feature value range is not constrained. For example, if the values of $f_{t,t}$ span all real-numbers, then it does not matter if you get $f_{t,n}$ by subtracting $f_{t,t}$ from $f_{t}$ as eq. (3), or by adding them as $f_{t,n}=f_{t}+f_{t,t}$ -- the Dense Block that outputs $f_{t,t}$ should be able to adjust its parameters to account for the opposite sign. Therefore, it is not clear if using the subtraction instead of addition is necessary.

- For the target and non-target feature plots in Figure 3, are they $f_t$ and $f_n$ extracted by the Conv 2D layer as eq. (1), or the refined features $\hat{f}_t$ and $\hat{f}_n$  before the IM-E (or IM-D)? Also, is it possible to compare $f_t$ to $\hat{f}_t$ and $f_n$ to $\hat{f}_n$? Knowing the difference will be helpful for understanding the contribution of the feature refinement step.

- In Figure 4 (a), the mask $M_t$ may be directly apply to extract the target features by multiplying it with the feature $\hat{f}_t$. So what is the role of the sigmoid here? (In other words, why should $|Sigmoid(\hat{f}_t)-M_t|\odot\hat{f}_t$ be considered instead of $M_t\odot\hat{f}_t$)?

- From Figure 9, it looks like that IM-E Modification-3 (d) is just taking off one Sigmoid operation from the Original IM-E (a). Then, why does the model size become smaller (1.64M) for (d) than the original size (1.74M) for (a) as presented in Table 8?

- On page 7, the paragraph below eq. (6): "... obtained by $f_n=f_4+f_{2,n}+f_{n,t}$" should be "$f_n=f_4+f_{2,n}+f_{3,n}$"?

---

> ### Author Response · Authors · 2024-11-23
> **Response to Reviewer S2co**
>
> Thank you so much for your valuable comments, which have been very helpful in improving our manuscript.
>
> Q1: The main weakness is there seems lacking a clear explanation as why the proposed architecture, especially the Local Modules (LMs), can promote target speech and non-target noise features extraction in the way as claimed in the paper. To be more specific, for the proposed LM-E (and LM-D) in Figure 2, what is the actual mechanism or reason that leads to the model assigning speech to the target branch and noise to the non-target branch? Please provide more detailed explanations on how such network design promotes speech and noise separation to carry out the proposed feature recycling scheme. Could the authors explicitly associate the "Network for speech extraction" in Figure 1 to the corresponding module(s) or layer(s) in Figure 2? This will help understanding where the recycling scheme actually takes place. Similarly, association of the "Network for noise extraction" in Figure 1 to the place(s) in Figure 2 will be helpful.
>
> A1: In LM, the extraction of non-target features through residuals is based on the principle of back-projection [1]. This method assumes that the information filtered out by the network primarily consists of dominant noise components, making it highly relevant for identifying non-target features. To ensure the effectiveness of this approach, normalization layers are applied after each convolutional module. These layers help preserve the energy range of the generated features, keeping them within the scale of the input features and preventing any significant deviation in overall output energy. This step is crucial, as uncontrolled variations in feature energy could distort the separation process, leading to undesired feature overlaps or loss of important information. By maintaining consistent energy distribution, the network can effectively isolate noise-like elements as residuals, which then serve as refined non-target features.
>
> GM utilizes energy distribution patterns in self-attention maps to extract non-target features. The attention mechanism divides the feature space into regions based on energy levels. High-energy areas represent target features, which are speech-dominant, while low-energy regions mainly contain noise-like elements. By measuring cross-similarity between these energy-based segments, the method ensures a clear separation of target and non-target features, avoiding any overlap of information. A key aspect of this approach is the consistent application of energy segmentation and feature extraction across the same spectral representation, which prevents unintended mixing of features. Additionally, the method does not rely on extra learnable parameters, ensuring that the non-target features are inherently noise-dominant, in line with the assumption that noise resides in low-energy regions
>
> We add the explanation of LM and GM for target and non-target feature extraction in Appendix A, while revising the content of Figure 2 and rewritten its caption to ensure that it aligns with the FRI strategy proposed in Figure 1.
>
> [1] Haris M, Shakhnarovich G, Ukita N. Deep back-projection networks for super-resolution. In Proceedings of the IEEE conference on computer vision and pattern recognition 2018 (pp. 1664-1673).

---

> ### Author Response · Authors · 2024-11-23
> **Response to Reviewer S2co**
>
> Q2: Another weakness of the paper is that the proposed SE model is tested on synthesized noisy speech data only. To better demonstrate the superiority of the proposed model, it will be nice if testing on real-world recorded noisy speech samples can be performed (e.g., using the realistic noisy samples from the CHiME-3 dataset: J. Barker et al., “The third ‘CHiME’ speech separation and recognition challenge: Analysis and outcomes,” Comput. Speech Lang, 2017). In this case, since there will be no ground-truth references to compute the objective metric scores used in the paper (i.e., PESQ, STOI, and SI-SNRi), the comparison with baselines may be done by using subjective listening testing or using automatic speech recognition (ASR) as evaluators.
>
> A2: We have presented added a comparison in terms of the WER metrics for FIRING-Net and the baseline models, evaluated on the same ASR system after processing the CHiME-3 real recorded test set, in Figure 10 and Appendix C.3.  The results are shown as follows:
>
> | Model        |  BUS |  CAF |  PED |  STR | Ave. |
> |--------------|:----:|:----:|:----:|:----:|:----:|
> | Noisy        | 51.8 | 34.7 | 27.2 | 20.1 | 33.4 |
> | PHASEN       | 22.4 | 14.1 | 13.5 | 11.2 | 13.8 |
> | SN-Net       | 16.8 | 16.3 | 14.2 | 14.0 | 15.1 |
> | Inter-SubNet | 23.5 | 14.1 | 11.2 |  9.7 | 14.9 |
> | CMGAN        | 15.2 | 13.3 | 12.4 |  9.9 | 12.3 |
> | MP-SENet     | 14.4 | 13.6 | 10.8 |  9.6 | 12.2 |
> | FIRING-Net   | 13.8 | 12.9 |  9.6 |  9.2 | 11.7 |
>
> Q3: In Figure 2, the LM-E (or LM-D) module has addition and subtraction operators. However, the subtraction operation may not lead to a different result from using addition if the feature value range is not constrained.
>
> A3: Before the subtraction operation, feature normalization is consistently applied. In the revised manuscript, we have validated the impact of normalization in Appendix C.5. Additionally, we added ablation experiments in the same section to evaluate the effect of normalization and the choice between addition and subtraction operations for the LM module , further demonstrating the effectiveness of our approach. The results are shown as follows:
>
> | Method                    | STOI   | PESQ | SI-SNRi |
> |---------------------------|--------|------|---------|
> | LM-E                      | 0.9135 | 3.03 | 13.96   |
> | _-w/o Norm + Subtraction_ | 0.8338 | 2.76 | 12.25   |
> | _-w. Norm + Addition_     | 0.8820 | 2.87 | 13.11   |
> | _-w/o Norm + Addition_    | 0.8454 | 2.78 | 12.18   |
> | LM-D                      | 0.9135 | 3.03 | 13.96   |
> | _-w.o Norm + Subtraction_ | 0.8557 | 2.81 | 12.54   |
> | _-w. Norm + Addition_     | 0.8739 | 2.85 | 12.89   |
> | _-w/o Norm + Addition_    | 0.8420 | 2.82 | 12.73   |
>
> The result indicating that when the normalization operation is removed, i.e., the features are unconstrained, there is little performance difference between subtraction and addition. However, when normalization is applied, constraining the features, the performance of the subtraction operation is significantly superior to that of the addition operation.

---

> > ### Author Response · Authors · 2024-11-23
> > **Response to Reviewer S2co**
> >
> > Q4: For the target and non-target feature plots in Figure 3, are they  f_t and  f_n extracted by the Conv 2D layer as eq. (1), or the refined features before the IM-E (or IM-D)? Also, is it possible to compare  coarse target and non-target features to refined  target and non-target features? Knowing the difference will be helpful for understanding the contribution of the feature refinement step.
> >
> > A4: We have revised Figure 3 to include the spectrograms of the Coarse target and non-target features. From these spectrograms, it can be observed that using only a single convolutional layer is insufficient for effectively distinguishing target and non-target features. Therefore, utilizing refined target and non-target features for subsequent processing is necessary.
> >
> > Q5: In Figure 4 (a), the mask M_t may be directly apply to extract the target features by multiplying it with the feature \hat{f}_t. So what is the role of the sigmoid here?
> >
> > A5: The use of Sigmoid aims to transition M_t from amplifying speech features to emphasizing noise features. This shift is essential because, at this stage, the non-target features, which inherently contain richer noise information, are used as guiding features. As epicted in Appendix B.4 (Internal structure disassembling for IM-E), directly applying M_t to amplify speech features might result in non-target features acting as interfering signals rather than providing a clear guiding direction. Such interference could significantly degrade performance. By contrast, the subtraction operation effectively utilizes M_t to enhance the noise characteristics, ensuring that the model leverages the noise-rich non-target features as a reliable source of guidance. This strategy leads to more effective extraction of target features and superior overall performance.
> >
> > Q6: From Figure 9, it looks like that IM-E Modification-3 (d) is just taking off one Sigmoid operation from the Original IM-E (a). Then, why does the model size become smaller (1.64M) for (d) than the original size (1.74M) for (a) as presented in Table 8? On page 7, the paragraph below eq. (6): "... obtained by
> >
> > A6: We are very sorry for this typo and has corrected. A careful proof-reading has been conducted to ensure no more such errors.

---

> ### Comment · Reviewer_S2co · 2024-11-24
> **Response to authors' rebuttal**
>
> I would like to thank the authors for providing detailed responses to my questions. My main concerns regarding LMs have been addressed by further clarification and added experimental results from the authors. The reviewer especially finds that the experiments on comparing addition and subtraction operations in the presence and absence of the normalization after the convolutional layer in LMs are very helpful (Figure 15 and Table 12). The revised manuscript has also improved the clarity and presentation of the main idea. Given the above, I increased my score from 5 to 6.
>
> Suggestion: Although the ASR comparison in Appendix C.3 is helpful, I think it would be nice to provide further details regarding the chosen ASR model.

---

> > ### Author Response · Authors · 2024-11-26
> > **Official Comment by Authors**
> >
> > Thank you for this kind suggestion. The ASR model we used is the official test model provided by CHiME-3 [1]. Specifically, it is based on the Kaldi recipe for Track 2 of the 2nd CHiME Challenge [2]. The DNN consists of 7 layers with 2048 units per hidden layer, and the input layer includes 5 frames of left and right context (i.e., 11×40 = 440 units). The DNN is trained using a standard procedure: pre-training with restricted Boltzmann machines, cross-entropy training, and sequence discriminative training with the state-level minimum Bayes risk (sMBR) criterion [3]. These details have been added to the revised Appendix C.3.
> >
> > [1] Barker J, Marxer R, Vincent E, Watanabe S. The third ‘CHiME’speech separation and recognition challenge: Dataset, task and baselines. In2015 IEEE Workshop on Automatic Speech Recognition and Understanding (ASRU) 2015 Dec 13 (pp. 504-511). IEEE.
> >
> > [2] C. Weng, D. Yu, S. Watanabe, and B.-H. F. Juang, “Recurrent deep neural networks for robust speech recognition,” in Proceedings of the 2014 IEEE International Conference on Acoustics, Speech, and Signal Processing (ICASSP), 2014, pp. 5532–5536.
> >
> > [3] K. Vesely, A. Ghoshal, L. Burget, and D. Povey, “Sequence-discriminative training of deep neural networks,” in Proceedings of the 14th Annual Conference of the International Speech Communication Association (INTERSPEECH 2013), 2013, pp. 2345–2349.

---

### Official Review · Reviewer_jpRr · 2024-11-01

**Soundness:** 2
**Presentation:** 3
**Contribution:** 3
**Rating:** 6
**Confidence:** 5

**Summary:**

The paper presents a novel approach to speech enhancement (SE) using a method called FIRING-Net, which incorporates a Filter-Recycle-Interguide (FRI) strategy. This strategy divides input features into target (speech) and non-target (noise) components, allowing for mutual refinement between these features.

**Strengths:**

1. The paper introduces a novel Filter-Recycle-Interguide (FRI) strategy for speech enhancement, which is a creative approach to refining speech and noise components. This method effectively separates and processes target and non-target features, offering a fresh perspective on enhancing speech quality.
2. The study conducted experiments on WSJ0-SI 84 + NOISEX-92 and AVSpeech + AudioSet. The results show substantial improvements in PESQ and SI-SNRi metrics compared to existing models. The paper also includes a comprehensive ablation study that validates the effectiveness of each component and the overall framework.

**Weaknesses:**

1. There are similar methods to this approach that also use cascaded strategies to enhance the model's ability to extract target speech, such as TaylorSENet [1]. However, these methods were not compared in the paper.

2. For the experimental section, the model trained on the DNS Challenge training set should naturally be tested on the public DNS Challenge test set. However, the authors chose two other datasets, which is quite odd. I believe the authors should test the model on the DNS Challenge test set to better demonstrate the model's performance.

3. In the appendix, the experimental results of VoiceBank-DEMAND show a significant discrepancy in the results of MP-SENet compared to the original paper. The authors mentioned that these results were not in the original paper, but they are detailed in the official repository: https://github.com/yxlu-0102/MP-SENet/blob/main/Figures/table.png. For detailed results, see: https://github.com/yxlu-0102/MP-SENet.

4. In the ablation study section, Table 3 is not referenced for the LM ablation study and should be added.

5. In the appendix, please specify whether the PESQ is narrowband or wideband, as these will yield different results.

6. The visualization results should include experiments with better-performing methods, such as MP-SENet, to see if similar issues persist.

7. For the experiments with IM-E Modification 2 and IM-D Modification 2, would incorporating both modules simultaneously yield better results? According to Tables 8 and 9, both modules perform second best but with fewer parameters, which is advantageous for speech enhancement. Is it possible to include both modules to reduce the parameter count and improve model performance?

8. Some dimension symbols in the paper, such as $L$, are not explained.

9. Word misuse: convolution operation -> convolutional operation

[1] Li A, You S, Yu G, et al. Taylor, can you hear me now? a Taylor-unfolding framework for monaural speech enhancement[J]. arXiv preprint arXiv:2205.00206, 2022.

**Questions:**

Consistent with the weaknesses

---

> ### Author Response · Authors · 2024-11-23
> **Response to Reviewer jpRr**
>
> We would like to truly thank for your valuable time and dedicated efforts to provide us with these constructive feedbacks.
>
> Q1: There are similar methods to this approach that also use cascaded strategies to enhance the model's ability to extract target speech, such as TaylorSENet [1]. However, these methods were not compared in the paper.
>
> A1: TaylorSENet remains focused on modeling the target speech by iteratively refining the spectrogram generated at the initial stage by calculating residuals to enhance its quality. To compare the performance of FIRING-Net and TaylorSENet, we added experiments on the publicly available DNS-Challenge dataset. The comparative results are presented in Table 7 of our revised Appendix C.1.
>
> |     Model   |   Feat.  |  Param.(M) |         | with Reverb |       |        |         | without Reverb |       |        |
> |:-----------:|:--------:|:----------:|:-------:|:-----------:|:-----:|:------:|:-------:|:--------------:|:-----:|:------:|
> |             |          |            | WB-PESQ |   NB-PESQ   |  STOI | SI-SDR | WB-PESQ |     NB-PESQ    |  STOI | SI-SDR |
> |    Noisy    |     -    |      -     |  1.822  |    2.753    | 86.62 |  9.033 |  1.582  |      2.454     | 91.52 |  9.070 |
> |   DCCRN-E   |    RI    |    3.70    |    -    |    3.077    |   -   |    -   |    -    |      3.266     |   -   |    -   |
> | Conv-TasNet | Waveform |    5.08    |  2.570  |      -      |   -   |    -   |  2.730  |        -       |   -   |    -   |
> |   PoCoNet   |    RI    |    50.00   |  2.832  |      -      |   -   |    -   |  2.748  |        -       |   -   |    -   |
> |    DCCRN+   |    RI    |    3.30    |    -    |    3.300    |   -   |    -   |    -    |      3.330     |   -   |    -   |
> |   TRU-Net   |    Mag   |    0.38    |  2.740  |    3.350    | 91.29 |  14.87 |  2.860  |      3.360     | 96.32 |  17.55 |
> |   CTS-Net   |  Mag+RI  |    4.99    |  3.020  |    3.470    | 92.70 |  15.58 |  2.940  |      3.420     | 96.66 |  17.99 |
> |  FullSubNet |    Mag   |    5.64    |  3.057  |    3.584    | 92.11 |  16.06 |  2.882  |      3.428     | 96.32 |  17.30 |
> | FullSubNet+ |  Mag+RI  |    8.67    |  3.177  |    3.648    | 93.64 |  16.44 |  3.002  |      3.503     | 96.67 |  18.00 |
> | TaylorSENet |    RI    |    5.40    |  3.330  |    3.650    | 93.99 |  17.10 |  3.220  |      3.590     | 97.36 |  19.15 |
> |    SICRN    |    RI    |    2.16    |  2.891  |    3.433    | 82.59 |  15.14 |  2.624  |      3.233     | 95.83 |  16.00 |
> |   CARNNHS   |    DFT   |      -     |  3.063  |    3.519    | 93.20 |  16.70 |  2.892  |      3.431     | 96.70 |  18.80 |
> |    MFNet    |    RI    |      -     |    -    |      -      |   -   |    -   |  3.430  |      3.740     | 97.87 |  20.31 |
> |  FIRING-Net |  Pha+Mag |    1.74    |  3.632  |    3.863    | 96.65 |  17.93 |  3.505  |      3.781     | 98.09 |  20.10 |
>
> Q2: For the experimental section, the model trained on the DNS Challenge training set should naturally be tested on the public DNS Challenge test set. However, the authors chose two other datasets, which is quite odd. I believe the authors should test the model on the DNS Challenge test set to better demonstrate the model's performance.
>
> A2: We initially did not conduct experiments on the VoiceBank+DEMAND dataset but not the DNS Challenge testing set since the results of most of the state-of-the-arts approaches can be obtained on the former dataset. We have added the model comparison on the DNS Challenge test set in Table 7 of our revised Appendix C.1. The comparison plase see above Table.

---

> > ### Author Response · Authors · 2024-11-23
> > **Response to Reviewer jpRr**
> >
> > Q3: In the appendix, the experimental results of VoiceBank-DEMAND show a significant discrepancy in the results of MP-SENet compared to the original paper. The authors mentioned that these results were not in the original paper, but they are detailed in the official repository: https://github.com/yxlu-0102/MP-SENet/blob/main/Figures/table.png. For detailed results, see: https://github.com/yxlu-0102/MP-SENet.
> >
> > A3: We adopted the metric results of MP-SENet from [1] since it is an official publication, and our FIRING-Net was designed based on this version of the model. Except for not utilizing MetricGAN, the loss functions and other settings we employed are consistent with this version. Therefore, to ensure a fair comparison, we used the results from this version of MP-SENet. Additionally, the version the reviewer mentioned incorporates STFT Consistency Loss and modifications in the bottleneck module [2], building upon the implementation in [1]. However, when MetricGAN is removed from [2] of MP-SENet, its performance is significantly lower than that of our FIRING-Net (MP-SENet [2]-w/o Metric loss: 3.44 PESQ; FIRING-Net: 3.57 PESQ).
> >
> > Q4: In the ablation study section, Table 3 is not referenced for the LM ablation study and should be added.
> >
> > A4: The citing of Table 3 has been added in the LM ablation study section.
> >
> > Q5: In the appendix, please specify whether the PESQ is narrowband or wideband, as these will yield different results.
> >
> > A5: The PESQ values in both the appendix and main manuscript are for wideband, and we have clarified this point in the revised version.
> >
> > Q6: The visualization results should include experiments with better-performing methods, such as MP-SENet, to see if similar issues persist.
> >
> > A6: In our revised Appendix, Figures 8 and 9 are added to show the enhanced spectrograms of CMGAN, MP-SENet, and FIRING-Net under -5 dB SNR under the Babble and Factory noise environments, respectively. The regions with the most noticeable differences are highlighted.
> >
> > Q7: For the experiments with IM-E Modification 2 and IM-D Modification 2, would incorporating both modules simultaneously yield better results? According to Tables 8 and 9, both modules perform second best but with fewer parameters, which is advantageous for speech enhancement. Is it possible to include both modules to reduce the parameter count and improve model performance?
> >
> > A7: While incorporating both IM-E Modification 2 and IM-D Modification 2 simultaneously may reduce parameters and yield better results than the ones of utilizing them individually, it is unlikely to yield better results than our current version of FIRING-Net. Each modification is optimized for its specific role in the encoder and decoder, with IM-E focusing on extracting speech from non-target features and suppressing noise, while IM-D refines discarded information to enhance reconstruction. Combining these simplified versions may disrupt the specialized roles and the mutual guidance mechanism, which is crucial to  our FRI framework, thereby compromising overall performance. Furthermore, the original IM-E and IM-D modules were designed to complement each other, ensuring robust feature refinement and noise suppression across diverse conditions. While the modified modules are more parameter-efficient individually, their combined use could result in trade-offs in scalability and robustness, which are vital for achieving SOTA performance in SE. Therefore, the original IM-E and IM-D modules remain the best choices for balancing performance and efficiency.
> >
> > Q8: Some dimension symbols in the paper, such as L, are not explained. Word misuse: convolution operation -> convolutional operation.
> >
> > A8: Thank you for identifying these issues. We have conducted a more thoroughcareful proof-reading of the manuscript to address these issues and correct the typos.
> >
> > [1] Lu, Y.-X., Ai, Y., Ling, Z.-H. (2023) MP-SENet: A Speech Enhancement Model with Parallel Denoising of Magnitude and Phase Spectra. Proc. INTERSPEECH 2023, 3834-3838, doi: 10.21437/Interspeech.2023-1441.
> >
> > [2] Lu YX, Ai Y, Ling ZH. Explicit estimation of magnitude and phase spectra in parallel for high-quality speech enhancement. arXiv preprint arXiv:2308.08926. 2023 Aug 17.

---

> > > ### Comment · Reviewer_jpRr · 2024-11-24
> > > **Additional verification of Q7**
> > >
> > > Thank you for the author's response, which has addressed most of my concerns. I hope the authors can validate the issue raised in **Q7** through experiments, as the behaviour of most deep-learning models often diverges from intuitive expectations. I have revised my score from 5 to 6. However, I encourage the authors to include this result in future iterations.

---

> > > > ### Author Response · Authors · 2024-11-26
> > > > **Official Comment by Authors**
> > > >
> > > > Thank you for this kind suggestion. Since there is not enough time to complete the experiments, we have provided a preliminary comparison of the model's performance at 30 epochs as a reference. The results are shown in the table below. We assure that we will include the full experimental comparison in the revised paper.
> > > >
> > > > | LM-E             | LM-D             |  PESQ  | STOI | SI-SNRi | Param.(M) |
> > > > |---------------------|---------------------|:------:|:----:|:-------:|:---------:|
> > > > | IM-E                | IM-D                | 0.8924 | 2.96 |  11.21  |    1.81   |
> > > > | IM-E Modification 2 | IM-D                | 0.8573 | 2.91 |  10.47  |    1.59   |
> > > > | IM-E                | IM-D Modification 2 | 0.8804 | 2.93 |  10.72  |    1.81   |
> > > > | IM-E Modification 2 | IM-D Modification 2 | 0.8513 | 2.88 |  10.51  |    1.59   |
> > > >
> > > > According to the results, it can be concluded that ombining IM-E Modification 2 and IM-D Modification 2 reduces the trainable parameters of the model, but the performance does not surpass that of the original settings of IM-E and IM-D.

---

### Official Review · Reviewer_hLf3 · 2024-11-04

**Soundness:** 2
**Presentation:** 2
**Contribution:** 2
**Rating:** 6
**Confidence:** 2

**Summary:**

This article presents FIRING-Net, a speech enhancement method based on the Filter-Recycle-Interguide strategy. This approach refines the extraction of speech and noise features through local and global modules, improving speech enhancement performance in complex noise environments. Experiments show that FIRING-Net significantly enhances performance across multiple datasets. The main contributions include a novel feature optimization strategy and experimental validation under various noise conditions.

**Strengths:**

1. The paper introduces a novel "Filter-Recycle-Interguide" (FRI) strategy, which is capable of effectively distinguishing between speech and noise features by mutual guidance, thereby achieving better noise reduction and speech restoration.

2. Through the design of local modules (LM) and global modules (GM), the paper effectively extracts target and non-target features by utilizing the energy distribution of convolutional module outputs and self-attention maps, respectively, thereby enhancing noise resistance performance.

3. By conducting extensive experiments on various public datasets and at different signal-to-noise ratio (SNR) levels, the proposed method demonstrates significant improvements in metrics such as PESQ and SI-SNRi compared to some existing methods, validating its effectiveness.

**Weaknesses:**

1. The paper's starting point is that "when noise and speech components are highly similar, it is difficult for the SE model to learn effective discrimination patterns." However, this aspect of the contribution is not prominently mentioned in the subsequent methodology analysis and experiments.

2. In the field of speech enhancement, there have been some related studies on refining and interacting features that were not referenced in the paper. Therefore, the selection of baseline methods for comparison in the experiments also lacks specificity.

3. There is a lack of qualitative analysis of the method and experiments, which results in a lack of interpretability and fails to distinguish the proposed method from existing research and the baseline methods used in the comparative experiments.

**Questions:**

1. Perform qualitative and interpretable analysis of the model method.
2. Explain how the subsequent experiments and the selection of baseline models for comparison reflect the challenges addressed by the paper’s starting point, which is “when noise and speech components are highly similar, it is difficult for the SE model to learn effective discrimination patterns.”

---

> ### Author Response · Authors · 2024-11-23
> **Response to Reviewer hLf3**
>
> We sincerely appreciate you for considering that our work is novel and effective. We respond your comments as follows:
>
> Q1: The paper's starting point is that "when noise and speech components are highly similar, it is difficult for the SE model to learn effective discrimination patterns." However, this aspect of the contribution is not prominently mentioned in the subsequent methodology analysis and experiments.
>
> A1: When background noise contains speech from other speakers or exhibits high energy that masks certain frequency bands of the speech signal, the inherent characteristics of speech may fail to distinguish themselves from noise. This often results in high similarity between speech and noise components, leading to speech distortion and residual noise [1]. To address this issue, this paper introduces the Filter-Recycle-Interguide (FRI) strategy, which recycles the information filtered by the network module and facilitates mutual guidance with the module's output. The recycled filtered information and the output are primarily dominated by noise and speech features, respectively, enabling mutual guidance to enhance their discriminability. This approach effectively increases the separability of noise and speech components from both feature perspectives, mitigating the issues of distortion and residual noise.
> In the experimental section, we evaluate the proposed FIRING-Net under challenging noise conditions using the AVSpeech + Audioset test set. Specifically, Babble and Baby Crying and Laughter noises are selected to test its performance under speech-interfering background noise, while Traffic and Engine noises are chosen to assess scenarios where high-intensity masking occurs in certain frequency bands of the target speech. The results presented in Table 2 demonstrate the superior performance of FIRING-Net under these noise conditions, further validating its effectiveness in addressing the aforementioned problem. To better illustrate this point, we have added the visualization of spectrograms, as shown in Figures 8 and 9, Appendix B.3, showing the enhanced speech under -5 dB conditions for CMGAN, MP-SENet, and FIRING-Net in babble noise and factory noise environments, to provide a more detailed comparison of its performance with other models under conditions where noise and speech components are highly similar.
>
> Q2: In the field of speech enhancement, there have been some related studies on refining and interacting features that were not referenced in the paper. Therefore, the selection of baseline methods for comparison in the experiments also lacks specificity. There is a lack of qualitative analysis of the method and experiments, which results in a lack of interpretability and fails to distinguish the proposed method from existing research and the baseline methods used in the comparative experiments.
>
> A2: There exist some related studies on interacting features, such as PHASEN [2], SN-Net [1] and Inter-SubNet [3]. These approaches have been adopted as the baseline methods, and our method differs from these approaches in that: 1) PHASEN is a model that employs an interactive framework for jointly modeling magnitude and phase information; 2) SN-Net adopts a dual-branch architecture to directly model the interaction between speech and noise features; 3) Inter-SubNet leverages subband interaction to explore cross-band dependencies; whereas CMGAN and MP-SENet, despite being state-of-the-art speech enhancement models, do not incorporate interactive feature modeling. FIRING-Net introduces a novel interaction mechanism between a network module's output and the residuals between its input and output. Compared to PHASEN, the features involved in FIRING-Net's interactions exhibit stronger inherent correlations than the magnitude and phase features used in PHASEN. In contrast to SN-Net, FIRING-Net effectively addresses the challenge of modeling noise features, which SN-Net fails to resolve. Furthermore, compared to InterSub-Net, FIRING-Net achieves a more comprehensive interaction by integrating both local and global feature perspectives. To more clearly present the experimental conclusions, we have revised and integrated this content, along with more detailed experimental analyses, into Section 5.1 of the main text.
>
> [1] Zheng C, Peng X, Zhang Y, Srinivasan S, Lu Y. Interactive speech and noise modeling for speech enhancement. In Proceedings of the AAAI Conference on Artificial Intelligence 2021 May 18 (Vol. 35, No. 16, pp. 14549-14557).
>
> [2] Yin D, Luo C, Xiong Z, Zeng W. Phasen: A phase-and-harmonics-aware speech enhancement network. InProceedings of the AAAI Conference on Artificial Intelligence 2020 Apr 3 (Vol. 34, No. 05, pp. 9458-9465).
>
> [3] Chen J, Rao W, Wang Z, Lin J, Wu Z, Wang Y, Shang S, Meng H. Inter-subnet: Speech enhancement with subband interaction. InICASSP 2023-2023 IEEE International Conference on Acoustics, Speech and Signal Processing (ICASSP) 2023 Jun 4 (pp. 1-5). IEEE.

---

> > ### Comment · Reviewer_hLf3 · 2024-11-25
> >
> > Thanks authors for the detailed response and I appreciate your hard work. Most of my concerns are well-addressed in the rebuttal. I raise my score from 5 to 6.

---

> > > ### Author Response · Authors · 2024-11-26
> > > **Official Comment by Authors**
> > >
> > > Thanks for the kind recognition of our work and support. We greatly appreciate the reviewer’s constructive comments, which helped us improve our work.

---

### Official Review · Reviewer_LH7g · 2024-11-06

**Soundness:** 4
**Presentation:** 4
**Contribution:** 3
**Rating:** 6
**Confidence:** 4

**Summary:**

Speech enhancement model with local and global information, where target and non-target portions interact with each other.

**Strengths:**

Idea appears to make sense, experimentation is done with care and results are good.

**Weaknesses:**

I am not completely sure that noise types used in training are not also seen in testing. What is clear is that SNR levels are same in training andf testing cases. So, it would be good to see how well the model generalizes to unseen conditions, unseen SNR level would be a fairly extreme tests case.

I also noticed that SNR scale is pretty extreme (from -5 to 5 db). How about using more moderate noise levels? Would the differences to competing models then disappear?

All ideas seem to be gathered from previous publications (like CV papers). It would be good to emphasize the novel technical contributions in the Introduction.

**Questions:**

- How do you know that in Eq. (1) convolution is going to separate target and non-target? In the Fig 3 it seems to be the case, but it is not clear to me that why that happens.
- Why in Table 6 lots of methods have same STOI value?
- In Eq. (14) some loss hyperparameters are explained, you set those to "0.2, 0.9, 0.1, and 0.3.", did you try to optimize these values? Even more importantly if you did experiment with these values, then did you look at the Table 1 and 2 while experimenting?  What I am looking for here is that how britle are these selections for practical use cases. Can someone just use them and assume that they will work out of the box?
-  Why reverb is used in the training, when testing no such cases are seen? How about removing reverb data from the training. Also, did you use the exactly same training set for all competing models?

---

> ### Author Response · Authors · 2024-11-23
> **Response to Reviewer LH7g**
>
> We would like to thank you for your efforts in reviewing our manuscript, and providing many helpful comments and suggestions, which will all prove invaluable in the revision and improvement of our paper.
>
> Q1: I am not completely sure that noise types used in training are not also seen in testing. What is clear is that SNR levels are same in training and testing cases. I also noticed that SNR scale is pretty extreme (from -5 to 5 db). How about using more moderate noise levels? Would the differences to competing models then disappear?
>
> A1: We confirm that the noise types used in the testing set of the AVSpeech+AudioSet dataset are not present in the training set. During training, we follow the existing setting [1] to draw noise samples from Audioset, DEMAND, and Freesound, and use AVSpeech + Audioset [2] for testing. The two common noise types “Babble” and “Traffic” that we have used in training, and the other two types “Engine” and “Baby Cry and Laughter” have not been observed. From the results in Table 2, we can see that our model can well generalize to new noise types. Regarding the generalization to unseen SNR levels, we have conducted additional experiments on the WSJ0-SI 84 + NOISEX-92 dataset at an unseen SNR level of -10 dB in two challenge noise environments, i.e., “Babble” and “Factory”. The results have been reported in Table 8 of our revised Appendix C.2,  which are summarized in the table below:
>
> |  Model       |  | Factory     |         |  |  Babble    |         |
> |--------------|:-------:|:----:|:-------:|:------:|:----:|:-------:|
> |              |   STOI  | PESQ | SI-SNRi |  STOI  | PESQ | SI-SNRi |
> | Noisy        |  0.4452 | 1.15 |    -    | 0.4338 | 1.07 |    -    |
> | PHASEN       |  0.5731 | 1.58 |  11.32  | 0.4882 | 1.39 |   8.58  |
> | SN-Net       |  0.6352 | 1.63 |  11.65  | 0.5011 | 1.45 |   9.21  |
> | Inter-SubNet |  0.6058 | 1.60 |  11.53  | 0.4945 | 1.46 |   9.14  |
> | CMGAN        |  0.6627 | 1.66 |  11.87  | 0.5259 | 1.51 |   9.86  |
> | MP-SENet     |  0.6584 | 1.68 |  12.24  | 0.5827 | 1.53 |  10.21  |
> | FIRING-Net   |  0.6986 | 1.75 |  13.12  | 0.6456 | 1.69 |  11.08  |
>
> and demonstrate that FIRING-Net maintains competitive performance even in such extreme conditions, highlighting its robustness under challenging scenarios. Moreover, we have demonstrated our model’s performance under more complex noise conditions, in Figure 6. This figure illustrates the performance of FIRING-Net in scenarios involving multiple noise types, indicating a more challenging and realistic test of the model’s generalization capabilities.
>
> Q2: All ideas seem to be gathered from previous publications (like CV papers). It would be good to emphasize the novel technical contributions in the Introduction.
>
> A2: We acknowledge that some inspirations for FIRING-Net, such as back-projection and Top-K mechanisms, originate from existing works, particularly from computer vision studies. However, one contribution of our FIRING-Net lies in its integration and tailored application of these ideas to the unique challenges of speech enhancement. The main contribution is that we propose a Filtering and Recycling (FRI) strategy, which is novel in leveraging the interaction between retained (target) and filtered-out (non-target) features. This strategy is specifically designed to address the unpredictable overlap between noise and speech, ensuring mutual refinement that enhances distinguishability. Moreover, the two-stage coarse-to-fine architecture in both local module (LM) and global module (GM) is a unique contribution that enables precise separation of target and non-target features. FIRING-Net further innovates by adapting interaction modules (IMs) differently in the encoder and decoder to accommodate the distinct feature compositions of these stages, enabling a robust process for noise elimination and speech reconstruction. These innovations are specifically crafted for speech enhancement, going beyond the direct adaptation of existing methods, and represent significant technical contributions to the field. We emphasized these points more clearly in the revised Introduction to highlight the originality of our approach.
>
> [1] Reddy CK, Dubey H, Gopal V, Cutler R, Braun S, Gamper H, Aichner R, Srinivasan S. ICASSP 2021 deep noise suppression challenge. InICASSP 2021-2021 IEEE International Conference on Acoustics, Speech and Signal Processing (ICASSP) 2021 Jun 6 (pp. 6623-6627). IEEE.
>
> [2] Ephrat A, Mosseri I, Lang O, Dekel T, Wilson K, Hassidim A, Freeman WT, Rubinstein M. Looking to listen at the cocktail party. ACM Transactions on Graphics. 2018 Jul 30;37(4):1-1.

---

> > ### Author Response · Authors · 2024-11-23
> > **Response to Reviewer LH7g**
> >
> > Q3: How do you know that in Eq. (1) convolution is going to separate target and non-target? In the Fig 3 it seems to be the case, but it is not clear to me that why that happens.
> >
> > A3: A single convolution operation is insufficient to effectively distinguish between target and non-target features. In the Local Module, we propose a two-stage coarse-to-fine strategy for feature separation. The first convolution layer performs only a coarse separation, serving as a preliminary step to segment features for subsequent refinement. The fine separation, expressed by Eq (2) and (3), is then achieved using a Dense Block, which further processes the coarse features to obtain the final target and non-target features. As illustrated in Figure 3 of the manuscript, the initial convolution-based coarse separation cannot accurately differentiate target and non-target features; it is the subsequent fine separation that achieves effective feature distinction.
> >
> > Q4: Why in Table 6 lots of methods have same STOI value?
> >
> > A4: The STOI values in Table 6 all appear to be the same (0.96) due to two reasons: (1) on the VoiceBank + DEMAND public dataset, STOI is typically reported to only two decimal places, and the values of compared methods in existing works are directly used in our paper. However, as observed in other tables within the document of our main manuscript, the STOI values often differ beyond the second decimal place (e.g., the third decimal place or further), which can not be reflected in Table 6, leading to an apparent uniformity in values; (2) STOI has an upper limit of 1.0, and achieving 0.96 is already a very high score, leaving little room for improvement. As a result, most models, including FIRING-Net and the compared approaches, achieve similarly high STOI values around 0.96 on this dataset.
> >
> > Q5: In Eq. (14) some loss hyperparameters are explained, you set those to "0.2, 0.9, 0.1, and 0.3.", did you try to optimize these values? Even more importantly if you did experiment with these values, then did you look at the Table 1 and 2 while experimenting? What I am looking for here is that how britle are these selections for practical use cases. Can someone just use them and assume that they will work out of the box?
> >
> > A5: The loss hyperparameters in Eq. (14) were set based on the configurations proposed in [3], which determined this set of parameters as the optimal balance through extensive experimentation. As also noted in [3], these values ensure that the magnitudes of all loss components are within a similar scale, preventing any single loss from dominating or being overshadowed during optimization. In future work, we plan to conduct more detailed experiments and analysis to derive a parameter configuration that not only achieves optimal performance but also provides a stronger interpretative foundation for practical applications.
> >
> > Q6: Why reverb is used in the training, when testing no such cases are seen? How about removing reverb data from the training. Also, did you use the exactly same training set for all competing models?
> >
> > A6: We ensured that all competing models were trained on the same training set and evaluated on the same testing set. In the WSJ0-SI 84 + NOISEX-92 testing set, reverb was randomly introduced during testing to match the level of reverb used in the training set. For the AVSpeech + AudioSet test set, no additional reverb was added, as the AVSpeech dataset inherently includes reverberant recordings. The inclusion of reverb during training is crucial because part of the reverberation in the mixed signal originates from the target speech itself, leading to significant feature differences between reverberation and noise. Models trained without reverb are less effective in handling reverberant signals during testing, as highlighted in [4]. In future work, we aim to address this limitation by investigating the inherent characteristics of reverberant signals to enhance the generalization capability of models, enabling them to better handle reverberant conditions.
> >
> > [3] Lu, Y.-X., Ai, Y., Ling, Z.-H. (2023) MP-SENet: A Speech Enhancement Model with Parallel Denoising of Magnitude and Phase Spectra. Proc. INTERSPEECH 2023, 3834-3838, doi: 10.21437/Interspeech.2023-1441.
> >
> > [4] Défossez, A., Synnaeve, G., Adi, Y. (2020) Real Time Speech Enhancement in the Waveform Domain. Proc. Interspeech 2020, 3291-3295, doi: 10.21437/Interspeech.2020-2409.

---

> > > ### Comment · Reviewer_LH7g · 2024-11-26
> > >
> > > I thank authors for their diligent answer to all my concerns. All my concerns are no satisfies now. I will keep my score as is (6).

---

### Author Response · Authors · 2024-11-23

Dear AC and Reviewers,

We sincerely thank all the reviewers for their time and thoughtful feedback. We appreciate the positive recognition of our work's novelty. In this section, we aim to clarify aspects of our paper and present additional results, which we will reference in our detailed responses to each reviewer. We will update the paper based on these discussions.

---

### Meta-Review · Area_Chair_41Fs · 2024-12-15

**Metareview:**

Five independent reviewers evaluated this work, which presents a speech enhancement method based on the Filter-Recycle-Interguide strategy. The proposed approach refines the extraction of speech and noise features using a combination of local and global blocks, demonstrating improved performance in challenging noisy environments.

Key strengths of the work include:
(i) the innovative idea of recycling intermediate speech and noise features to enhance the separation of target and non-target components, and
(ii) a robust experimental evaluation.

However, the proposed method lacks a solid theoretical explanation for why the chosen operations yield the observed results. To their credit, the authors made an effort to address this concern during the discussion phase. While the approach builds on existing ideas, it introduces sufficient originality to distinguish itself.

Overall, the work contributes novel elements to the field, and its performance evaluations under both synthetic and real noisy conditions are convincing.

The submission lacks experiments without reverberant speech in the training process, as requested by Reviewer LH7g.

**Additional Comments On Reviewer Discussion:**

The discussion phase was highly constructive, with the authors addressing key concerns raised by the reviewers. The main points of feedback included:

(i) Insufficient experimental evidence on real-world data.
(ii) Limited clarity regarding certain technical aspects, particularly the role of Eq. (1).
(iii) A lack of qualitative analysis of the method and experiments, which affected interpretability and made it difficult to differentiate the proposed method from existing research.
(iv) An unclear comparison with related approaches, as some ideas in the work appeared to overlap with existing literature.
(v) An absence of theoretical justification for the operations used in the proposed architecture.
(vi) Unclear usage of reverberation in training (not presente  in testing)

The authors thoroughly addressed the first three concerns and supplemented their work with new experimental results. In response to (iv), they clarified that the novelty lies in the innovative and effective application of existing concepts. Additionally, the authors provided a reasonable explanation regarding the role of the operations in their architecture. Unfortunately, the authors' answer given to point (vi) is not fully convincing. Nonetheless, the authors clarified that the same experimental conditions were used for proposed and competing models.

The changes made after the discussion phase primarily include: (i) the inclusion of new experimental results, (ii) an improved discussion highlighting the differences from related works and clarifying the novel aspects of the proposed approach, and (iii) a clearer explanation of the role of the operations within the proposed architecture.

---

### Decision · Program_Chairs · 2025-01-22

Accept (Poster)